# Beyond Heuristic Tuning: Power-Calibrated LLM Watermarking

**Xiaopu Wang** [1]  **Zelin He** [1]  **Chengyuan Liu** [1]  **Runze Li** [1]

## Abstract

Logit-based watermarking is a widely used mechanism for identifying LLM generated content, yet its effectiveness is governed by a fundamental trade-off between detectability and semantic distortion. Existing analyses provide limited guidance for principled hyperparameter selection, leaving practical deployments reliant on heuristic tuning. In this work, we develop a power-calibrated statistical framework that establishes explicit quantitative relationships between watermark hyperparameters, detection power, and distortion. This characterization transforms watermark design into a guided optimization problem. Building on these results, we derive practical parameter selection procedures that achieve optimal trade-offs under constraints. Extensive experiments across multiple language models and datasets validate the theory and demonstrate that the proposed framework consistently identifies Pareto-optimal points.

## 1. Introduction

Large language models (LLMs) have achieved remarkable progress in natural language processing, enabling high performance across a wide range of tasks including reasoning, programming, and creative writing (Achiam et al., 2023; Liu et al., 2024; Team et al., 2024). Concerns have been raised regarding potential misuse (Shumailov et al., 2023). One mitigation strategy is to enable identification of machine-generated text. However, recent work has shown that post-hoc detection methods that distinguish LLM-generated text from human text after generation are fundamentally fragile (Mitchell et al., 2023; Sadasivan et al., 2023). These limitations motivate proactive approaches that embed signals directly into generation.

Text watermarking provides a scalable mechanism for establishing the provenance of language-model-generated content. Broadly, watermarking schemes introduce controlled statistical dependencies between generated text and a secret key (Kuditipudi et al., 2023; Li et al., 2025; Kirchenbauer et al., 2024a). Among these, logit-based watermarking has emerged as a practical and widely adopted paradigm. These methods introduce small, structured perturbations to the model's logits, which subtly bias the sampling distribution toward pseudo-random token subsets, preserving the native sampling process while enabling efficient statistical detection.

This paper focuses on logit-based watermarking and addresses a central unresolved challenge: *a precise quantification of the trade-off between detectability and distortion.* Stronger logit perturbations improve detection while simultaneously inducing larger distributional shifts, degrading semantic fidelity. Recent work has characterized this trade-off under different objectives, including log-perplexity analysis (Wouters, 2024) and KL-constrained formulations (Cai et al., 2024). However, these frameworks provide limited theoretical guidance for parameter calibration, leaving current systems reliant on heuristic tuning without reliable performance guarantees across deployment settings.

In this work, we address this limitation by developing a power-calibrated statistical framework. Our approach establishes explicit quantitative relationships between watermark hyperparameters, detectability, and distortion. This quantification transforms watermark design from heuristic trial-and-error into a guided optimization problem. Specifically, our contributions can be summarized as follows:

- **Theoretical Characterization** We develop a framework that yields *closed-form relationships* linking watermark hyperparameters to detection power and distortion, providing a characterization of the detectability-distortion trade-off.
- **Principled Parameter Selection** We derive procedures for selecting watermark parameters that achieve optimal points under power or distortion constraints, reducing hyperparameter tuning to a tractable optimization problem.
- **Empirical Validation** We validate our theoretical predictions in multiple language models and datasets, demonstrating that the proposed framework consistently identifies Pareto-optimal configurations and outperforms heuristic pa-

---
[1]Department of Statistics, Pennsylvania State University, University Park, PA, USA. Correspondence to: Xiaopu Wang <xmw5221@psu.edu>.

*Proceedings of the 43rd International Conference on Machine Learning*, Seoul, South Korea. PMLR 306, 2026. Copyright 2026 by the author(s).

rameter selection strategies in practice.

## 2. Preliminary

**Notation** Let $V$ denote the vocabulary, with size $|V|$. For a generated token sequence $(x_1, \ldots, x_n)$, we denote the prefix up to position $i$ by $x_{<i} := (x_1, \ldots, x_{i-1})$. At each step $i$, the base language model induces a distribution $P_i$ over $V$, where $p_i(k) = \mathbb{P}(x_i = k \mid x_{<i})$. Watermarking modifies this distribution into $Q_i$, with probabilities $q_i(k) = \mathbb{P}_\delta(x_i = k \mid x_{<i})$, where $\mathbb{P}_\delta$ denotes the probability measure under watermarking.

**Logit-Based Watermark** We focus on the logit-based KGW watermarking framework (Kirchenbauer et al., 2024a), which has been widely studied in this line of work (Cai et al., 2024; Kirchenbauer et al., 2024b; Wouters, 2024; Zhao et al., 2023). At each generation step $i$, the vocabulary $V$ is pseudo-randomly partitioned into a *green list* $G_i \subset V$ of size $\gamma|V|$ and its complement $R_i := V \setminus G_i$. Watermark embedding is performed by adding a constant bias $\delta > 0$ to the logits of green-list tokens. The two parameters $(\gamma, \delta)$ govern the process jointly. Let $l_{i,k}$ denote the original logit of token $k$ at step $i$. The modified logits are

$$l'_{i,k} = \begin{cases} l_{i,k} + \delta, & k \in G_i, \\ l_{i,k}, & k \in R_i. \end{cases}$$

which yields the watermarked distribution $Q_i$. Detection is then formulated as a hypothesis test comparing the observed frequency of green-list tokens against the null hypothesis $H_0$ corresponding to non-watermarked generation.

**Detectability-Distortion Trade-Off** An effective watermark must balance two competing objectives:

1. **Detectability:** the reliability with which the watermark can be identified from generated text.

2. **Distortion:** the deviation of the watermarked output distribution from the original model behavior.

In the KGW framework, such a trade-off is governed jointly by $(\gamma, \delta)$: increasing $\delta$ strengthens the bias and improves detectability, but simultaneously increases distortion, while $\gamma$ controls the fraction of tokens affected by the bias. Our goal is to explicitly characterize this trade-off by deriving quantitative relationships that link $(\gamma, \delta)$ to detectability and distortion, thereby transforming watermark design from ad hoc parameter tuning into a theoretically grounded optimization problem.

### 2.1. Related Work and Existing Gap

A principled optimization of watermarking strategies requires metrics that faithfully capture both semantic fidelity

and detection reliability. We start with a review of related work on metric selection and outline the existing gap.

**Distortion** To quantify distortion, we seek a metric that reflects the extent to which the watermark alters the original model distribution. Prior work (Wouters, 2024) quantifies semantic degradation via the shift in log perplexity, $\tilde{\mathbb{E}}[\log \text{PPL}] - \mathbb{E}[\log \text{PPL}]$, where perplexity is defined as:

$$\text{PPL}(x_{1:n}) = \exp\left( -\frac{1}{n} \sum_{t=1}^n \log P(x_t \mid x_{<t}) \right).$$

While (Wouters, 2024) argue this metric is optimal under independence and fixed $\gamma$, the independence assumption rarely holds and optimal $\gamma$ remains unknown. Furthermore, Cai et al. (2024) demonstrate that this metric favors deterministic generation, which compromises output diversity. Alternatively, they propose using the KL divergence between the watermarked and unwatermarked distributions. For a generation step $i$, this is defined as:

$$D_{\text{KL}}(Q_i \parallel P_i) = \sum_{v \in V} q_i(v) \log \frac{q_i(v)}{p_i(v)}.$$

While KL divergence offers a principled information-theoretic measure of the loss induced by watermarking, Cai et al. (2024) rely on plug-in estimates without formal theoretical justification. *Consequently, it remains unclear whether these estimates accurately reflect the true divergence in watermarking scenarios.*

**Detectability** For detectability, our goal is to measure how reliably a watermarked sequence can be distinguished. Wouters (2024) propose using the expected green-token count difference under the watermarked distribution, given by:

$$\tilde{\mathbb{E}}[\Delta N_G], \quad N_G := \sum_{i=1}^n \mathbf{1}_{x_i \in G_i}$$

While Cai et al. (2024) analyze the per-step probability shift

$$DG(i) = \sum_{k \in G_i} q_i(k) - \sum_{k \in G_i} p_i(k).$$

The primary difference between the two lies in the level of aggregation: the former accumulates the discrepancy, while the latter captures the per-step value. An increased discrepancy intuitively facilitates detection. However, this heuristic does not necessarily provide an accurate characterization of detectability. From a statistical perspective, detectability is best defined in terms of *statistical power*, where rejecting $H_0$ corresponds to declaring the sample watermarked. *The remaining gap lies in identifying a proper representation for direct calculation of statistical power.*

# 3. A Formal Statistical Framework

We start with formalizing the hypothesis testing problem.

## 3.1. Null Hypothesis

**Quantifying the Green-List Mechanism**   In the KGW generation process, a green list is generated via a hash function applied to the generation context. Hash functions are commonly modeled as pseudo-random functions that produce approximately source-independent outputs that are uniformly distributed over their range (Bellare and Rogaway, 1993). We formalize this idea in the following assumption:

**Assumption 3.1** (Random green-list assignments)**.** The green lists $\{G_i\}_{i=1}^n$ are i.i.d. distributed over the vocabulary, and are independent of the generated token sequence $(x_1, \ldots, x_n)$.

Assumption 3.1 isolates the randomness introduced by the watermarking mechanism and separates it from the intrinsic randomness of language model generation. Under Assumption 3.1, we have the following result:

**Lemma 3.2.** *Let $I_i := \mathbf{1}\{x_i \in G_i\}$. Under Assumption 3.1, the indicators $\{I_i\}_{i=1}^n$ are independent and identically distributed as*
$$I_i \sim \mathrm{Bernoulli}(\gamma).$$

*Proof.* See Appendix A.1.                                    □

Therefore, for the sequence-level statistic
$$S_n = \sum_{i=1}^n I_i,$$

under $H_0$, $S_n$ admits the following central-limit normal approximation:
$$Z_n^{(0)} := \frac{S_n - n\gamma}{\sqrt{n\gamma(1-\gamma)}} \xrightarrow{d} \mathcal{N}(0,1).$$

**Watermark Detection as Hypothesis Testing**   The KGW framework increases the likelihood of sampling green-list tokens. Let $\mathbb{P}_0$ denote the unwatermarked generation distribution and let $\mathbb{P}_\delta$ denote the watermarked distribution induced by the logit shift $\delta$. Concretely, $\mathbb{P}_0(x_i = k \mid x_{<i})$ is the original next-token distribution of the language model, while $\mathbb{P}_\delta(x_i = k \mid x_{<i})$ is the distribution after adding the green-list logit bias. Since $G_i$ is determined by the context and secret key, this notation suppresses the conditioning on $G_i$ when no confusion is possible. Although the watermark perturbs the next-token distribution at each generation step, detection is performed at the sequence level. The token-level characterization is

$$H_0: \ \mathbb{P}_0(x_i \in G_i) = \gamma, \qquad H_1: \ \mathbb{P}_\delta(x_i \in G_i) > \gamma.$$

**Remark.**   The original null statement, "text is not watermarked," is composite: it may include human-written text and outputs from different unwatermarked models. Since the distribution of human text is generally unavailable, this null is difficult to specify directly as a distributional hypothesis on $V^n$. The KGW formulation avoids modeling these heterogeneous source distributions and instead uses the keyed green-list signal. Under Assumption 3.1, any source that does not use the secret key and watermarking rule has green-list membership probability $\gamma$; for unwatermarked model outputs this gives $\mathbb{P}_0(x_i \in G_i) = \gamma$. Under the watermarked distribution, the logit bias increases the same probability to $\gamma'(\gamma, \delta) > \gamma$, as characterized below. Thus $I_i = \mathbf{1}\{x_i \in G_i\}$ records the KGW signal observed by the keyed detector, and $S_n = \sum_{i=1}^n I_i$ aggregates this signal over the sequence.

The actual detector is the sequence-level one-sided test

$$\frac{S_n - n\gamma}{\sqrt{n\gamma(1-\gamma)}} > z_{1-\alpha}.$$

However, a key challenge is that the alternative hypothesis is only partially specified at this stage. To derive a principled expression for detection power, it is necessary to explicitly model the effect of watermarking on the token distribution under the alternative.

## 3.2. Token-Level Alternative

**Token-Level Effect**   We first characterize how the KGW watermarking mechanism alters the probability of generating green-list tokens at the token level.

Let $\boldsymbol{\sigma}^{(i)} \in \Delta^{|V|-1}$ denote the next-token probability (NTP) vector at position $i$, where $V$ is the vocabulary and $\Delta^{|V|-1} = \{p \in \mathbb{R}^V \mid p_k \geq 0, \ \sum_{k \in V} p_k = 1\}$ denotes the probability simplex. The following lemma provides an explicit relationship between the green-token probability under the unwatermarked and watermarked generation processes.

**Lemma 3.3.** *Conditioned on the NTP vector $\boldsymbol{\sigma}^{(i)}$, we have*

$$\mathbb{P}_\delta(x_i \in G_i \mid \boldsymbol{\sigma}^{(i)}) = \frac{e^\delta \, \mathbb{P}(x_i \in G_i \mid \boldsymbol{\sigma}^{(i)})}{(e^\delta - 1) \, \mathbb{P}(x_i \in G_i \mid \boldsymbol{\sigma}^{(i)}) + 1}.$$

*Proof.* See Appendix A.2.                                    □

Lemma 3.3 provides a precise, token-level description of how watermarking biases generation toward the green list. Next, we obtain a sequence-level quantity by accounting for the distribution of the NTP vectors $\boldsymbol{\sigma}^{(i)}$.

**NTP Distribution**   A precise characterization of the NTP vector $\boldsymbol{\sigma}^{(i)}$ induced by a large language model is generally intractable, due to its dependence on high-dimensional internal representations (Farquhar et al., 2024). To enable

a tractable yet principled analysis, we introduce a surrogate probabilistic model for $\boldsymbol{\sigma}^{(i)}$. We begin by specifying a baseline probabilistic assumption on the NTP distribution.

**Assumption 3.4** (Non-informative NTP prior). At each position $i$, the NTP vector $\boldsymbol{\sigma}^{(i)} \in \Delta^{|V|-1}$ is modeled as an independent draw from the uniform Dirichlet distribution,

$$\boldsymbol{\sigma}^{(i)} \sim \text{Dir}(1, \ldots, 1),$$

Assumption 3.4 is a non-informative, model- and text-agnostic prior for the NTP vector. It does not use corpus-specific frequency or semantic information, matching the fair comparison setting in which the baseline watermarking methods also do not use such information. The framework is not tied to this prior; when reliable structural information is available, richer priors such as mixtures of Dirichlet (Dalal and Misra, 2024) or Latent Dirichlet allocation (Blei et al., 2003) can be used. See Appendix A.4 for a detailed discussion.

The approximation is effective because the detector depends on the green-list mass $Y_i = \sum_{v \in G_i} \sigma_v^{(i)}$, not the full NTP vector. Since the keyed green list is sampled independently of the language model, $\mathbb{E}(Y_i \mid \boldsymbol{\sigma}^{(i)}) = \gamma$; with a large vocabulary, $Y_i$ concentrates near $\gamma$, and sequence-level averaging further reduces fluctuations. We then derive the corresponding green-token probability under watermarking.

**Theorem 3.5** (Green-token probability under non-informative prior). *Under Assumption 3.4, the marginal probability of sampling a green-list token under watermarking satisfies*

$$\gamma' := \mathbb{P}_\delta(x \in G)$$
$$= e^\delta \gamma \, {}_2F_1\big(1, \, \gamma|V| + 1; \, |V| + 1; \, -(e^\delta - 1)\big),$$

*where ${}_2F_1(\cdot)$ denotes the Gaussian hypergeometric function. Moreover, in the large-vocabulary regime,*

$$\gamma' = \frac{e^\delta \gamma}{1 + \gamma(e^\delta - 1)} + O(|V|^{-1/2}).$$

*Proof.* See Appendix A.4. $\qquad\qquad\square$

Theorem 3.5 plays a central role in the subsequent analysis. With this characterization in place, the token-wise hypothesis testing problem can now be transformed into

$$H_0 : \mathbb{P}(x_i \in G_i) = \gamma, \qquad H_1 : \mathbb{P}_\delta(x_i \in G_i) = \gamma'.$$

It replaces the originally implicit and model-dependent alternative hypothesis with an explicit, low-dimensional parameterization in terms of $\gamma'$. In Section 5, we empirically verify that this approximation closely matches observed green-token rates across multiple language models and datasets, even for moderate vocabulary sizes.

### 3.3. Sequence-Level Alternative

We now consider the aggregate statistic $S_n$ under the alternative. Unlike the null case, the sequence of indicators $I_t$ is no longer independent due to the autoregressive nature of the generation process.

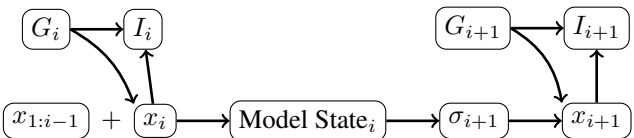

*Figure 1.* Information flow between tokens, model state, and green lists.

**Dependence Structure and Information Flow** We investigate the extent to which the future indicators $X_t^+ = (I_i)_{i \geq t}$ depend on the past indicators $X_t^- = (I_i)_{i < t}$. We first dissect the dependence pathways. As illustrated in Figure 1, the realization of an indicator $I_t$ depends on two factors: the generated token $x_t$ and the green list partition $G_t$. Crucially, the green list $G_t$ is essentially unobservable to the model. Thus, $G_t$ is statistically orthogonal to the model. The only pathway for dependence between indicators is the semantic trajectory through correlation with tokens:

$$I_s \leftarrow x_s \rightarrow \text{Model State} \rightarrow x_t \rightarrow I_t, \quad \text{for } s < t.$$

Since the language model is trained to optimize semantic coherence rather than preserve arbitrary historical noise, information about past specific realizations of $I_i$ is compressed and overwritten. We model this "forgetting" mechanism using decay in mutual information:

$$\mathcal{I}(X; Y) = H(X) - H(X \mid Y).$$

where $H(X)$ denotes the entropy of the random variable $X$. Since the model parameters remain fixed during generation, the governing mechanism does not change across steps. The rate of decay should therefore be constant. The generation process of $G_i$ is also invariant across time. This motivates the following assumption.

**Assumption 3.6** (Information Decay). We assume that there exist constants $C > 0$ and a decay rate $0 < \rho < 1$ such that for any $s < t$:

$$\mathcal{I}(X_t^+; X_s^-) \leq C\rho^{t-s}.$$

Also $\{I_t\}$ is assumed to be stationary.

To establish the theoretical results, we further preclude some pathological cases where negative correlations perfectly cancel out the external randomness of $G_t$ in the long run.

**Assumption 3.7** (Non-degenerate Long-run Variance). The variance of the indicator sequence is strictly positive asymptotically:

$$\sigma^2 := \lim_{n \to \infty} \frac{1}{n} \operatorname{Var}\left(\sum_{i=1}^n I_i\right) > 0.$$

Next, we are ready to establish the asymptotic distribution of the aggregate statistics under the alternative:

**Theorem 3.8.** *Under Assumptions 3.6 and 3.7, suppose that under the alternative* $\mathbb{E}I_t = \gamma'$, *where* $0 < \gamma' < 1$. *Then, for some constant* $c > 0$,

$$\sqrt{n}\left(\frac{S_n}{n} - \gamma'\right) \xrightarrow{d} \mathcal{N}(0, c\gamma'(1 - \gamma')).$$

*Proof.* See Appendix A.3. □

For the level-$\alpha$ test above, define the detection power as

$$\pi^*(\gamma, \delta) := \mathbb{P}_\delta(\text{reject } H_0).$$

Combining the central-limit approximation under $H_0$, the rejection threshold under $H_0$, and the asymptotic distribution under $H_1$ in Theorem 3.8 yields the following normal-approximation power:

$$\pi^*(\gamma, \delta) \approx \Phi\left(\frac{\sqrt{n}(\gamma' - \gamma) - z_{1-\alpha}\sqrt{\gamma(1 - \gamma)}}{\sqrt{c\gamma'(1 - \gamma')}}\right). \quad (1)$$

**Remark.** The power depends not only on the green-token ratio gap $\gamma' - \gamma$, but also on the values of $\gamma$ and $\gamma'$. This demonstrates that differences in green-token counts or probabilities alone are insufficient for measuring power. The constant $c > 0$ captures long-run variance inflation caused by dependence in generated text. It changes the numerical value of the predicted power, but not the parameter that maximizes the normal-approximation objective. To see this, write the objective in the form

$$\Phi\left(\frac{f(\gamma, \delta)}{\sqrt{c}}\right),$$

where $f$ is independent of $c$. Since $\Phi$ is monotone and $1/\sqrt{c}$ is a positive constant, maximizing this expression is equivalent to maximizing $f(\gamma, \delta)$. Thus the parameter selection is independent of $c$. Appendix C discusses how $c$ can be estimated in practice.

### 3.4. Distortion Metrics

**KL Divergence**   The other essential part of our framework is the distortion metric. As stated in Cai et al. (2024), the total KL divergence of the sequence is equal to the sum of token-wise KL divergences. The watermark-induced distortion is therefore measured using the expected token-wise KL divergence $D_{\text{KL}}(\gamma, \delta)$, with direction from the watermarked distribution to the original distribution.

**Lemma 3.9** (Plug-in Formula for KL Divergence). *Under the large-vocabulary plug-in approximation* $P(x \in G) \approx \gamma$, *let*

$$\gamma' = \frac{e^\delta \gamma}{1 + \gamma(e^\delta - 1)}$$

*be the corresponding approximation to the effective green-token probability. Then*

$$D_{\text{KL}}(\gamma, \delta) = \delta\gamma'(\gamma, \delta) - \log\left(1 + \gamma(e^\delta - 1)\right),$$

*Given* $\gamma$, $D_{\text{KL}}(\gamma, \delta)$ *is strictly increasing in* $\delta$ *for all* $\delta > 0$.

$$\frac{\partial}{\partial \delta} D_{\text{KL}}(\gamma, \delta) > 0.$$

*As* $\delta$ *varies and* $\gamma$ *fixed,* $D_{\text{KL}}(\gamma, \delta)$ *is upper bounded by the choice of* $\gamma$

$$\sup_\delta D_{\text{KL}}(\gamma, \delta) = -\log\gamma.$$

*As* $\gamma$ *varies and* $\delta$ *fixed,* $D_{\text{KL}}(\gamma, \delta)$ *attains maximum at*

$$\gamma_0(\delta) = \frac{\delta e^\delta - (e^\delta - 1)}{(e^\delta - 1)^2}.$$

*Proof.* See Appendix A.5. □

Lemma 3.9 implies that for any fixed $\gamma$, the strict monotonicity of $D_{\text{KL}}(\gamma, \delta)$ in $\delta$ establishes a one-to-one correspondence between the logit shift $\delta$ and the induced distortion level. This allows watermark strength to be parameterized directly in terms of an explicit distortion budget rather than an abstract model parameter. As a consequence, watermark design under distortion constraints can be formulated as a principled optimization problem, where $\delta$ is determined implicitly by the target KL divergence.

## 4. Practical Guidance

We now translate our theoretical analysis into concrete guidance for practical watermark parameter selection. In contrast to prior work, which treats the watermark parameters $(\gamma, \delta)$ as unconstrained hyperparameters tuned heuristically on downstream data, our framework casts watermark design as a theoretically informed optimization problem to balance *detectability* and *distortion*.

**From Hyperparameter Tuning to Optimization**   Our results provide explicit quantitative characterizations of the two competing objectives in watermark design:

- **Detectability**: captured by the asymptotic power function $\pi^*(\gamma, \delta)$ in Eq. (1),

- **Distortion**: measured by the expected token-wise KL divergence $D_{\mathrm{KL}}(\gamma, \delta)$ in Eq. (3.9).

A key consequence of Lemma 3.9 is that, for any fixed $\gamma$, the KL divergence is strictly increasing in $\delta$ and admits a closed-form expression. This establishes a one-to-one correspondence between the logit shift $\delta$ and the induced distortion level $K$, allowing watermark strength to be parameterized directly in terms of distortion rather than an abstract model parameter. As a result, the original two-dimensional hyperparameter tuning problem over $(\gamma, \delta)$ is reduced to a one-dimensional numerical optimization problem. For instance, given a distortion budget $K_0$, watermark design can be formulated as

$$\max_{\gamma} \ \pi^*(\gamma, \delta(\gamma, K_0)). \tag{2}$$

where $\delta(\gamma, K_0)$ is obtained by solving $D_{\mathrm{KL}}(\gamma, \delta) = K_0$. This optimization is solved once before generation, not separately at each token: both $\pi^*(\gamma, \delta)$ and $D_{\mathrm{KL}}(\gamma, \delta)$ are expectation-level functions of the watermark parameters, rather than functions of a realized token or context. Once the calibrated pair $(\gamma^*, \delta^*)$ is selected, the same parameters are applied uniformly throughout the generated sequence. Analogously, one may minimize distortion subject to a target detectability constraint. In both cases, parameter selection is guided by the theory, and we later show that the resulting optimization is efficient and stable in practice.

**Initializing the Green-List Rate $\gamma$**   For solving problem (2), what remains is to provide principled guidance on how to initialize the numerical search on $\gamma$.

We begin by noting a structural property of the distortion constraint. For a fixed watermark strength parameter $\delta_0 > 0$, the function $D_{\mathrm{KL}}(\gamma, \delta_0)$ is unimodal in $\gamma$: it increases for small $\gamma$, attains a maximum at an intermediate value, and then decreases as $\gamma$ approaches one. Consequently, for a fixed distortion budget $K_0$,

$$D_{\mathrm{KL}}(\gamma, \delta_0) = K_0. \tag{3}$$

admits at most two solutions, denoted by $\gamma_\ell < \gamma_h$, whenever a solution exists. To determine which solution is preferable, we examine the corresponding detection power.

Figure 2(a) reports the statistical power achieved by the two solutions $\gamma_\ell$ and $\gamma_h$ as the distortion budget $K_0$ varies. At higher power levels, the higher solution $\gamma_h$ consistently dominates, yielding a strictly lower distortion level with equivalent power. Figure 2(b) shows that the power at the intersection point where the two solutions coincide is uniformly below $0.8$ for all tested values of $\delta$. This implies that for practically relevant target power levels (e.g., $0.8$ or higher), the higher solution $\gamma_h$ is inherently optimal,

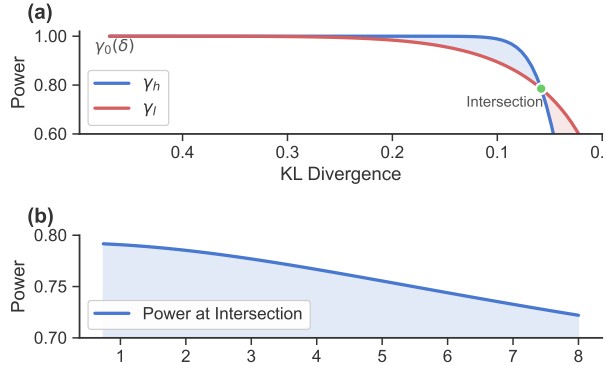

*Figure 2.* Figure (a) illustrates the power curves of the two solutions $\gamma_l, \gamma_h$ of Eq. 3, calculated with $\alpha = 0.05$, $\delta = 2$, and $n = 50$. The two curves intersect at one point. At power levels above the intersection, $\gamma_h$ achieves equivalent power with lower $D_{\mathrm{KL}}$. Figure (b) extends the calculation to a wide range of $\delta \in (0.5, 8)$ by plotting the power at the corresponding intersection point. Each $\delta$ corresponds to one intersection point. We observe a monotonically decreasing pattern, with maximum power below $0.8$.

requiring substantially less distortion to reach the target detectability. We verify this fact with experiments showing that values of $\gamma$ below $\gamma_0$ are non-optimal; see Appendix E.

Further notice that under the normal approximation, requiring the detector to outperform random guessing (power $> 0.5$) imposes an explicit upper bound $\gamma \leq \gamma^*$ (see Appendix A.6 for details). Moreover, with $\gamma_h > \gamma_0$, $\gamma_h$ lies in a region where $D_{\mathrm{KL}}$ is monotone decreasing with respect to $\gamma$. Therefore, for target power levels above $0.8$, the optimal solution is guaranteed to lie just below the bound $\gamma^*$.

Accordingly, we initialize the numerical search for $\gamma$ in a neighborhood immediately below $\gamma^*$ and search downward. This strategy reliably recovers the larger root $\gamma_h$, which is optimal for practically relevant power targets.

## 5. Experiments

### 5.1. Experimental Setup

Our experimental protocol largely follows prior watermarking evaluations (Kirchenbauer et al., 2024b). The implementation is available at https://github.com/shooof/wm. We evaluate across multiple pretrained language models, including OPT (Zhang et al., 2022), Pythia (Biderman et al., 2023), and GPT-2 (Radford et al., 2019). To assess robustness across domains, we consider text from the Colossal Clean Crawled Corpus (C4) (Raffel et al., 2023), Long-Form Question Answering (LFQA) (Xu et al., 2023), and Wikipedia (Foundation). All models and datasets are obtained from Hugging Face. For each sample, we use the first 50 tokens as a prompt and generate $n = 50$ tokens unless stated otherwise. Detection is performed on these short

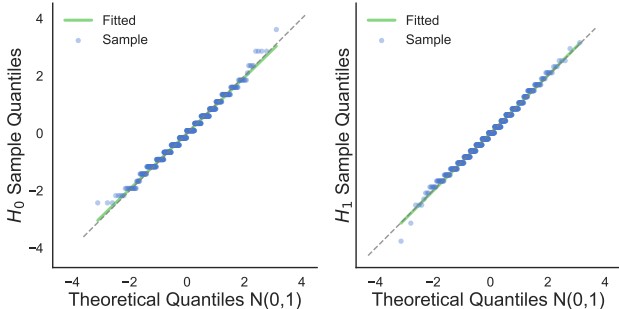

**Figure 3.** Q–Q plots of the standardized green-token count statistic under both hypotheses. The empirical quantiles of the statistic are compared with the theoretical quantiles of a standard normal distribution. Left: $H_0$, corresponding to unwatermarked text. Right: $H_1$, corresponding to watermarked text.

generations to avoid trivial detectability gains from signal accumulation in long contexts and to better expose differences in statistical efficiency across methods (Kirchenbauer et al., 2024b).

We compare our method against the framework (Cai et al., 2024) (DP in our figures), OPT watermarking (Wouters, 2024), and a dense grid-search baseline representing heuristic hyperparameter tuning. All experiments are conducted using the codebase of Kirchenbauer et al. (2024b), with additional implementation of KL divergence computation. For the search, DP, and our proposed method, identical execution pipelines are used to control for implementation effects, ensuring that performance differences arise solely from parameter selection. Detectability is evaluated using the true positive rate (TPR) at a fixed significance level $\alpha = 0.05$, based on a normalized $z$-score detector with threshold $z_{1-\alpha}$. Semantic distortion is quantified using the per-token KL divergence from the watermarked next-token distribution to the unwatermarked next-token distribution (computed from logits), averaged over the generated sequence. Details on hyperparameter ranges and additional implementation specifics are provided in Appendix B.

### 5.2. Distributional Verification

We now empirically validate the distributional assumptions underlying our theoretical analysis.

In particular, Lemma 3.2 and Theorem 3.8 establish that, under mild conditions, the standardized green-token count statistic converges to a standard normal distribution under both the null and alternative hypotheses.

Figure 3 depicts Q–Q plots of the standardized green-token count statistic. Under both $H_0$ (unwatermarked text) and $H_1$ (watermarked text), the empirical quantiles closely track the theoretical quantiles of a standard normal distribution. This agreement holds across the distribution, with only minor tail

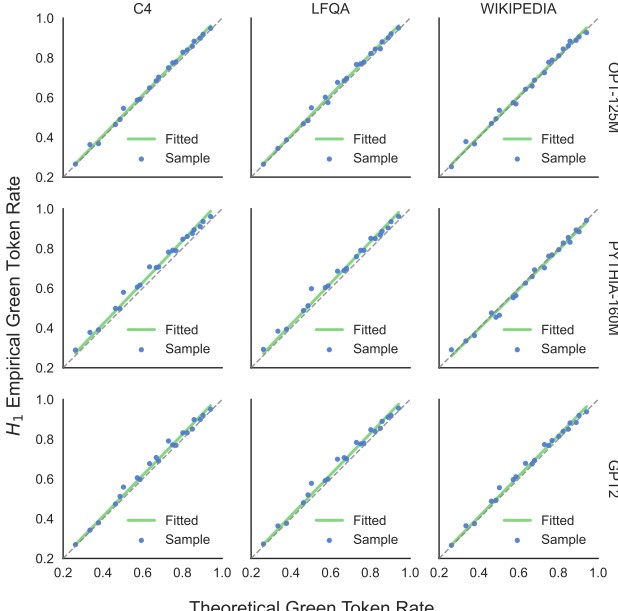

**Figure 4.** Comparison between theoretical and empirical green-token rates. The fitted model is $y = kx$, tested across the C4, LFQA, and Wikipedia datasets.

deviations attributable to finite-sample effects.

These results provide empirical support for the normal approximation used in our detectability analysis and justify the use of the $z$-score detector throughout our experiments. In particular, they confirm that the power expressions derived in Section 3 accurately capture the behavior of the test statistic in practical regimes.

### 5.3. Verification of the Alternative Hypothesis Characterization

Next, we evaluate the accuracy of our theoretical characterization of the alternative hypothesis. Theorem 3.8 provides an explicit mapping between the watermark parameters $(\gamma, \delta)$ and the marginal green-token probability $\gamma'$.

Figure 4 compares the theoretical green-token rate predicted by Theorem 3.8 with the empirically observed rate across a grid of watermark parameters, language models, and datasets. Across all settings, the empirical measurements exhibit a near-linear relationship with the theoretical predictions, with coefficients of determination exceeding $R^2 \geq 0.98$ in all cases. This close alignment confirms that the simplified modeling assumptions introduced in Section 3 used to derive $\gamma'$ are sufficient to accurately describe the watermarking mechanism in practice.

A complete table of $R^2$ values for the linear fits of green-token rate and $D_{\mathrm{KL}}$ is included in Appendix B.9.

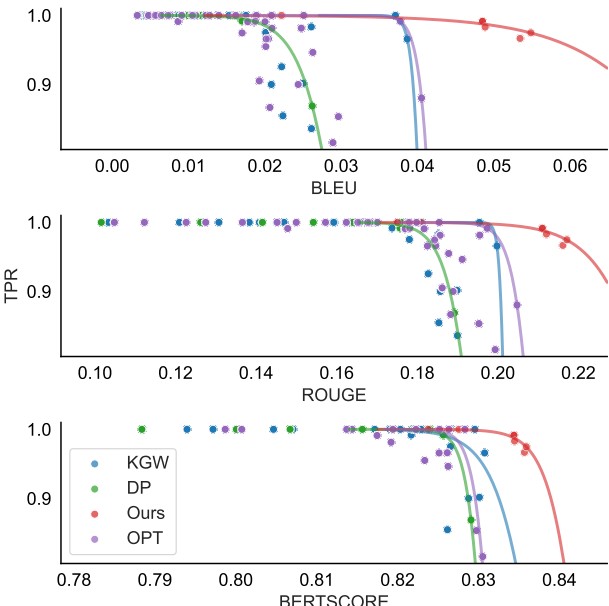

*Figure 5.* Statistical power (measured by TPR) versus semantic metrics, including BLEU, ROUGE, and BERTScore. Higher values indicate better quality for all metrics. Results are obtained on the LFQA dataset. For each method, the curve is fitted using its Pareto frontier with a small tolerance allowed for visualization quality. The fitted curve follows the form $y = 1/(1 + e^{ax-b})$.

## 5.4. Performance

We evaluate the empirical detectability-distortion trade-off of different watermarking strategies by measuring statistical power (TPR) versus semantic distortion (token-wise KL divergence). We use $-\log(D_{\text{KL}})$ for better visualization across our experiments. Figure 6 reports results on all three datasets (C4, LFQA, and Wikipedia) using GPT-2, OPT-125M, and Pythia-160M. To assess whether the observed pattern persists for a larger model, Figure 7 in the appendix reports the same evaluation on Gemma-2 9B.

Across these models, our method consistently lies on the Pareto frontier, achieving high detection power at substantially lower distortion compared to baseline methods. In particular, for moderate distortion regimes (larger $-\log(D_{\text{KL}})$), our approach maintains near-saturated TPR, while KGW and DP baselines exhibit an earlier and steeper degradation in power. This behavior is especially pronounced for OPT-125M and Pythia-160M, where heuristic baselines incur a sharp drop in TPR once distortion is reduced beyond a model-dependent threshold.

Notably, the fitted Pareto curves reveal a systematic separation between methods: OPT-based and heuristic strategies operate in suboptimal regions of the distortion-detectability space, often requiring significantly higher KL divergence to achieve comparable power. In contrast, our method exhibits a more favorable slope, indicating improved statistical efficiency in converting distortion budget into detectability gains. This aligns with our theoretical analysis, which predicts that principled parameter selection based on explicit power-distortion relationships yields superior operating points.

Overall, these results demonstrate that the proposed framework not only improves peak detectability, but more importantly, achieves robust detection performance under tight distortion constraints. This validates the practical effectiveness of replacing heuristic hyperparameter tuning with theoretically informed optimization, as developed in Sections 3 and 4.

**Evaluation Under Alternative Quality Metrics.** Optimizing watermark parameters with respect to a single quality metric may bias the solution toward specific surface-level properties of the generated text. To assess the robustness of our approach, we therefore evaluate detectability-distortion trade-offs under multiple complementary semantic quality metrics, including BLEU (Papineni et al., 2002), ROUGE (Lin, 2004), and BERTScore (Zhang et al., 2020).

Figure 5 depicts a representative comparison of statistical power (TPR) versus text quality measured by these metrics on the LFQA dataset. The complete panels across all three datasets and models, together with regression summaries relating KL distortion to sequence-level quality, are provided in Appendix B.13 and Figure 8. Across all metrics, we observe a consistent qualitative pattern: our method maintains near-saturated detection power over a substantially wider range of quality values compared to baseline approaches, while heuristic methods exhibit a sharper degradation in TPR once quality constraints become stringent.

Although BLEU, ROUGE, and BERTScore emphasize different aspects of text quality, the relative ordering of watermarking methods remains stable. In particular, KGW and DP baselines tend to sacrifice semantic fidelity more aggressively to achieve high power, leading to early collapses in TPR when quality thresholds are tightened. In contrast, our method exhibits a smoother Pareto frontier across all metrics, indicating that the optimized parameterization induces a more uniform and controlled distributional shift.

This consistency across metrics suggests that the advantage of our approach is not tied to a specific notion of text similarity, but rather reflects a fundamentally more efficient use of distortion budget. By explicitly optimizing watermark parameters through a statistically grounded framework, the induced perturbation aligns more closely with the intrinsic geometry of the model's output distribution, resulting in robust detectability without disproportionate degradation under either surface-level or semantic evaluation criteria.

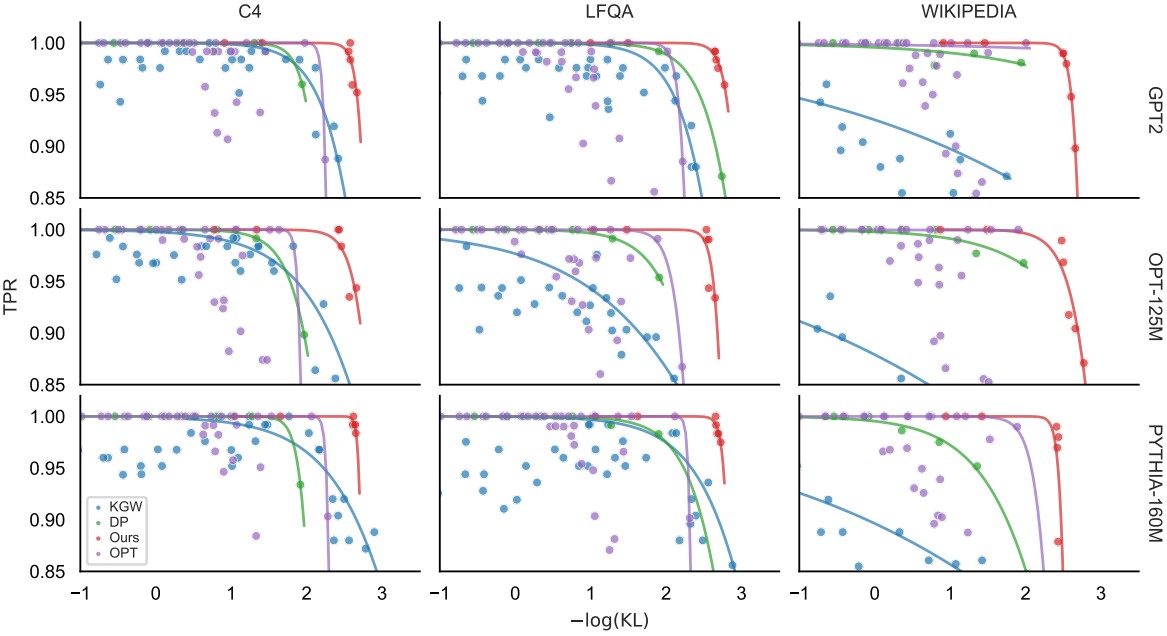

*Figure 6.* Statistical power (measured by TPR) versus semantic distortion (measured by KL divergence). Results are obtained on all three datasets using GPT-2, OPT-125M, and Pythia-160M. For each method, the curve is fitted using its Pareto frontier with a small tolerance allowed for visualization quality. The fitted curve follows the form $y = 1/(1 + e^{ax-b})$.

Overall, these results confirm that the proposed watermarking strategy achieves genuinely low distortion while preserving strong detection performance across a broad range of quality measures, reinforcing the practical value of theoretically informed parameter optimization.

## 6. Conclusion

We introduced a controllable statistical framework for logit-based watermarking that enables principled calibration of watermark strength under explicit detectability and distortion objectives. By establishing quantitative mappings between watermark parameters, detection power, and KL-based distortion, our approach transforms watermark design from heuristic tuning into statistically grounded optimization. These findings provide a practical foundation for deploying watermarking systems with reliable statistical guarantees. A discussion of the limitations is provided in Appendix D.

## Acknowledgements

This research was supported a NSF grant DMS 2514400 and two NIH grants R01GM163244 and AI192205. The content is solely the responsibility of the authors and does not necessarily represent the official views of NSF or NIH.

## Impact Statement

Text watermarking has significant societal impact by helping distinguish human-written content from machine-generated text in an era of large-scale automated content production. It supports the fight against misinformation, academic dishonesty, and malicious uses of generative AI. Our work improves the efficiency of such techniques, fostering a more scalable and practical deployment of watermarking in real-world systems.

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

# A. Proof and Discussion of Lemmas and Theorems

### A.1. Proof of Lemma 3.2

For $a \in \{0, 1\}$, define

$$p(a) := \gamma^a (1 - \gamma)^{1-a}.$$

Let

$$\mathcal{X} := \sigma(x_1, \ldots, x_n)$$

be the sigma-field generated by the token sequence. Under Assumption 3.1, the green lists $G_1, \ldots, G_n$ are independent of $\mathcal{X}$ and are mutually independent. Moreover, for any fixed token $v \in V$,

$$\mathbb{P}(v \in G_i) = \gamma.$$

Thus, conditional on $\mathcal{X}$, the token $x_i$ is fixed and

$$\mathbb{P}(I_i = a \mid \mathcal{X}) = p(a), \qquad a \in \{0, 1\}.$$

We now prove by induction that for every $k = 1, \ldots, n$ and every $(a_1, \ldots, a_k) \in \{0, 1\}^k$,

$$\mathbb{P}(I_1 = a_1, \ldots, I_k = a_k \mid \mathcal{X}) = \prod_{i=1}^{k} p(a_i).$$

The case $k = 1$ follows from the preceding display. Suppose the statement holds for $k - 1$. Since $I_1, \ldots, I_{k-1}$ are measurable with respect to $\sigma(\mathcal{X}, G_1, \ldots, G_{k-1})$, and since $G_k$ is independent of $\mathcal{X}, G_1, \ldots, G_{k-1}$,

$$\mathbb{P}(I_k = a_k \mid \mathcal{X}, G_1, \ldots, G_{k-1}) = p(a_k).$$

Therefore, by the tower property,

$$
\begin{aligned}
&\mathbb{P}(I_1 = a_1, \ldots, I_k = a_k \mid \mathcal{X}) \\
&= \mathbb{E}[\mathbf{1}\{I_1 = a_1, \ldots, I_{k-1} = a_{k-1}\} \mathbb{P}(I_k = a_k \mid \mathcal{X}, G_1, \ldots, G_{k-1}) \mid \mathcal{X}] \\
&= p(a_k) \, \mathbb{P}(I_1 = a_1, \ldots, I_{k-1} = a_{k-1} \mid \mathcal{X}) \\
&= \prod_{i=1}^{k} p(a_i).
\end{aligned}
$$

This completes the induction. Taking expectations over $\mathcal{X}$ gives

$$\mathbb{P}(I_1 = a_1, \ldots, I_n = a_n) = \prod_{i=1}^{n} p(a_i) = \prod_{i=1}^{n} \mathbb{P}(I_i = a_i).$$

Hence $I_1, \ldots, I_n$ are mutually independent. Since each $I_i \sim \text{Bernoulli}(\gamma)$, the sequence $\{I_i\}_{i=1}^{n}$ is i.i.d. Bernoulli($\gamma$).

### A.2. Proof of Lemma 3.3

We first review the token generation process of LLMs and introduce the necessary preliminaries.

Let the softmax mapping $\boldsymbol{\sigma} : \mathbb{R}^{|V|} \to \Delta^{|V|-1}$ be defined as

$$\boldsymbol{\sigma}_v(\boldsymbol{z}) = \frac{e^{z_v}}{\sum_{u \in V} e^{z_u}}, \qquad v \in V,$$

where $\Delta^{|V|-1} = \left\{ \boldsymbol{p} \in \mathbb{R}_{\geq 0}^{|V|} \mid \sum_{v \in V} p_v = 1 \right\}$ denotes the probability simplex over the vocabulary $V$.

Given a prefix $x_{<i}$, let $M(x_{<i}) \in \mathbb{R}^{|V|}$ denote the logit vector produced by the language model. The next token is sampled according to

$$\mathbb{P}(x_i = v \mid x_{<i}) = \boldsymbol{\sigma}_v(M(x_{<i})), \qquad x_i \sim \text{Categorical}(\boldsymbol{\sigma}(M(x_{<i}))).$$

**Green-List Probability.** Let $G_i \subseteq V$ denote the green list at position $i$. The probability that the generated token belongs to the green list is

$$\mathbb{P}(x_i \in G_i \mid x_{<i}) = \sum_{v \in G_i} \mathbb{P}(x_i = v \mid x_{<i}) = \sum_{v \in G_i} \boldsymbol{\sigma}_v\big(M(x_{<i})\big).$$

Define

$$A = \sum_{v \in G_i} e^{M_v(x_{<i})}, \qquad B = \sum_{v \notin G_i} e^{M_v(x_{<i})}.$$

Then

$$\mathbb{P}(x_i \in G_i \mid x_{<i}) = \frac{A}{A + B}.$$

Equivalently, letting

$$R = \frac{B}{A},$$

we have

$$\mathbb{P}(x_i \in G_i \mid x_{<i}) = \frac{1}{1 + R}.$$

**Logit Biasing and Watermarked Distribution.** To embed a watermark, KGW adds a constant bias $\delta > 0$ to the logits corresponding to tokens in the green list. Specifically, for each $v \in V$,

$$\mathbb{P}_\delta(x_i = v \mid x_{<i}) = \boldsymbol{\sigma}_v\big(M(x_{<i}) + \delta\,\mathbf{1}_{v \in G_i}\big).$$

The resulting green list probability under the biased distribution is

$$\mathbb{P}_\delta(x_i \in G_i \mid x_{<i}) = \sum_{v \in G_i} \mathbb{P}_\delta(x_i = v \mid x_{<i}) = \frac{e^\delta A}{e^\delta A + B} = \frac{1}{1 + \frac{R}{e^\delta}}.$$

**Closed-Form Relationship.** Let

$$p := \mathbb{P}(x_i \in G_i \mid x_{<i})$$

Since $R = (1 - p)/p$, we obtain

$$\mathbb{P}_\delta(x_i \in G_i \mid x_{<i}) = \frac{e^\delta}{e^\delta + \frac{1-p}{p}} = \frac{e^\delta\,p}{(e^\delta - 1)p + 1}$$

Therefore, the green list probability after watermark embedding satisfies

$$\mathbb{P}_\delta(x_i \in G_i \mid x_{<i}) = \frac{e^\delta\,\mathbb{P}(x_i \in G_i \mid x_{<i})}{(e^\delta - 1)\mathbb{P}(x_i \in G_i \mid x_{<i}) + 1}$$

Since the conditional distribution of $x_i$ given $x_{<i}$ is categorical with parameter $\boldsymbol{\sigma}(M(x_{<i}))$, conditioning on $x_{<i}$ is equivalent to conditioning on the induced distribution parameter.

$$\mathbb{P}_\delta(x_i \in G_i \mid \boldsymbol{\sigma}^{(i)}) = \frac{e^\delta\mathbb{P}(x_i \in G_i \mid \boldsymbol{\sigma}^{(i)})}{(e^\delta - 1)\mathbb{P}(x_i \in G_i \mid \boldsymbol{\sigma}^{(i)}) + 1}.$$

### A.3. Proof of Theorem 3.8

*Proof.* The sequence is strictly stationary with marginal distribution $I_t \sim \text{Bernoulli}(\gamma')$. For integers $a, b$, define

$$\mathcal{F}^b_{-\infty} := \sigma(I_t : t \le b), \qquad \mathcal{F}^\infty_a := \sigma(I_t : t \ge a).$$

Essentially, Assumption 3.6 is:

$$\mathcal{I}(\mathcal{F}^b_{-\infty}; \mathcal{F}^\infty_a) \le C\rho^{a-b}, \qquad a > b.$$

For two sub-$\sigma$-algebras $\mathcal{U}, \mathcal{V} \subset \mathcal{A}$, define alpha mixing

$$\alpha(\mathcal{U}, \mathcal{V}) := \sup_{A \in \mathcal{U},\, B \in \mathcal{V}} \left| \mathbb{P}(A \cap B) - \mathbb{P}(A)\mathbb{P}(B) \right|$$

For $n \geq 1$, define the strong mixing coefficients of $\{I_t\}$ by

$$\alpha(n) := \sup_{j \in \mathbb{Z}^+} \alpha\left( \mathcal{F}_{-\infty}^j, \mathcal{F}_{j+n}^\infty \right)$$

Let $\mathcal{I}(\mathcal{U}; \mathcal{V})$ denote the coefficient of information between $\sigma$-fields $\mathcal{U}$ and $\mathcal{V}$. By Assumption 3.6 there exist constants $C > 0$ and $\rho \in (0, 1)$ such that, for all $j \in \mathbb{Z}$ and all $n \geq 1$,

$$\mathcal{I}\left( \mathcal{F}_{-\infty}^j; \mathcal{F}_{j+n}^\infty \right) \leq C\rho^n \tag{4}$$

By (Bradley, 2005), Eqs. (1.11) and (1.18),

$$2\alpha(\mathcal{U}, \mathcal{V}) \leq \left[ \mathcal{I}(\mathcal{U}; \mathcal{V}) \right]^{1/2}$$

Applying this with $\mathcal{U} = \mathcal{F}_{-\infty}^j$ and $\mathcal{V} = \mathcal{F}_{j+n}^\infty$ gives, for all $j \in \mathbb{Z}$,

$$\alpha\left( \mathcal{F}_{-\infty}^j, \mathcal{F}_{j+n}^\infty \right) \leq \tfrac{1}{2} \left[ \mathcal{I}\left( \mathcal{F}_{-\infty}^j; \mathcal{F}_{j+n}^\infty \right) \right]^{1/2}$$

Taking the supremum over $j \in \mathbb{Z}$ and using (4) yields

$$\alpha(n) \leq \tfrac{1}{2}\sqrt{C} \left( \sqrt{\rho} \right)^n.$$

Therefore, $\{I_t\}$ is geometrically $\alpha$-mixing.

**CLT** We have $\mathbb{E}|I_1|^{2+\delta} < \infty$ for $\delta = 1$, and

Since $\alpha(n)$ decays geometrically, we have

$$\sum_{n=1}^\infty \alpha(n)^{\delta/(2+\delta)} < \infty$$

along with Assumption 3.7, by the CLT for $\alpha$-mixing sequences (Doukhan, 1995), condition (3) on p. 45,

$$\frac{\sum_{t=1}^n I_t - n\mathbb{E}I_t}{\sqrt{n}} \xrightarrow{d} \mathcal{N}(0, \sigma^2),$$

where

$$\sigma^2 = c\,\mathrm{Var}(I_1) = c\gamma'(1 - \gamma').$$

Therefore,

$$\sqrt{n} \left( \frac{S_n}{n} - \gamma' \right) \xrightarrow{d} \mathcal{N}(0, c\gamma'(1 - \gamma')).$$

$\square$

### A.4. Proof of Theorem 3.5

We begin by proving the theorem with a fixed subset $G \subseteq V$.

Assume that

$$\boldsymbol{\sigma} \sim \mathrm{Dir}(1, \ldots, 1),$$

and let $|V|$ denote the vocabulary size.

Our goal is to compute

$$\mathbb{P}(x \in G) = \mathbb{E}_{\sigma}\left[\sum_{i \in G} \sigma_i\right], \qquad \mathbb{P}_{\delta}(x \in G) = \mathbb{E}_{\sigma}\left[\frac{e^{\delta}}{e^{\delta} + \left(\sum_{i \in G} \sigma_i\right)^{-1} - 1}\right]$$

Define

$$Y := \sum_{i \in G} \sigma_i$$

By the additivity property of the Dirichlet distribution, we have

$$Y \sim \text{Beta}(\alpha_G, \alpha_{G^c}), \qquad \alpha_G = \sum_{i \in G} 1 = |G|, \quad \alpha_{G^c} = \sum_{i \in G^c} 1 = |V| - |G|$$

Using the properties of the Beta distribution,

$$\mathbb{P}(x \in G) = \mathbb{E}[Y] = \frac{\alpha_G}{\alpha_G + \alpha_{G^c}} = \frac{|G|}{|V|} = \gamma.$$

By definition,

$$\mathbb{P}_{\delta}(x \in G) = \mathbb{E}\left[\frac{e^{\delta}}{e^{\delta} + Y^{-1} - 1}\right] = \mathbb{E}\left[\frac{e^{\delta}Y}{1 + (e^{\delta} - 1)Y}\right]$$

Using the beta density and denoting the beta function by $B(a, b)$,

$$\mathbb{P}_{\delta}(x \in G) = \frac{e^{\delta}}{B(|G|, |V| - |G|)} \int_0^1 \frac{y}{1 + (e^{\delta} - 1)y} y^{|G|-1}(1 - y)^{|V|-|G|-1} \, dy$$

Equivalently, this expression admits the closed form

$$\mathbb{P}_{\delta}(x \in G) = e^{\delta}\frac{|G|}{|V|} \, {}_2F_1\big(1, |G| + 1; |V| + 1; -(e^{\delta} - 1)\big) = e^{\delta}\gamma \, {}_2F_1\big(1, \gamma|V| + 1; |V| + 1; -(e^{\delta} - 1)\big),$$

where ${}_2F_1$ is the Gauss hypergeometric function. This gives the exact evaluation of the probability. We then show the asymptotic limit.

**Asymptotic Limit.** Recall,

$$Y \sim \text{Beta}(|G|, |V| - |G|),$$

Consequently,

$$\mathbb{E}[Y] = \gamma, \qquad \text{Var}(Y) = \frac{|G|(|V| - |G|)}{|V|^2(|V| + 1)} \sim \frac{1}{|V|}$$

Since $\text{Var}(Y) \to 0$, we have $Y \xrightarrow{P} \gamma$.

The function

$$g(y) = \frac{e^{\delta}y}{1 + (e^{\delta} - 1)y}$$

is continuous and bounded on $[0, 1]$, hence by the dominated convergence theorem,

$$\mathbb{E}[g(Y)] \to g(\gamma) = \frac{e^{\delta}\gamma}{1 + \gamma(e^{\delta} - 1)}$$

We compute

$$g'(y) = \frac{e^{\delta}}{(1 + (e^{\delta} - 1)y)^2} \leq e^{\delta}, \qquad y \in [0, 1],$$

Thus, $g$ is $e^\delta$-Lipschitz:

$$|g(y) - g(\gamma)| \leq e^\delta |y - \gamma|$$

Taking expectation and applying Jensen,

$$|\mathbb{E}[g(Y)] - g(\gamma)| \leq e^\delta \mathbb{E}|Y - \gamma| \leq e^\delta \sqrt{\operatorname{Var}(Y)} \sim O(|V|^{-1/2})$$

Thus

$$\mathbb{P}_\delta(x \in G) = \frac{e^\delta \gamma}{1 + \gamma(e^\delta - 1)} + O(|V|^{-1/2}).$$

**Averaging over Random $G$.** If $G$ is drawn uniformly, then in the symmetric Dirichlet setting the quantities $\mathbb{P}(x \in G)$ and $\mathbb{P}_\delta(x \in G)$ depend only on $|G|$ and not on the specific realization of $G$. Consequently,

$$\mathbb{E}_G[\mathbb{P}(x \in G)] = \gamma, \qquad \mathbb{E}_G[\mathbb{P}_\delta(x \in G)] = \mathbb{P}_\delta(x \in G),$$

and the above limit remains valid after averaging over $G$.

**Discussion of the General Case.** One may argue that token distributions in natural language are inherently uneven. Mixtures of Dirichlet distributions (Dalal and Misra, 2024) and latent Dirichlet models (Blei et al., 2003) naturally accommodate imbalanced token frequencies.

For general Dirichlet parameters $(\alpha_1, \ldots, \alpha_{|V|})$, the aggregated parameter $\alpha_G = \sum_{i \in G} \alpha_i$ becomes random when $G$ is sampled uniformly. When the vocabulary size is large and the Dirichlet parameters are not excessively heterogeneous, standard concentration results for sampling without replacement imply that $\alpha_G/\alpha_0$ concentrates around $\gamma$. In this setting, the symmetric Dirichlet analysis derived above provides an accurate approximation to the general case, with deviations vanishing as the total concentration parameter increases.

**Numerical Verification.** As a sanity check, we numerically evaluated the exact finite-$|V|$ expression and compared it to its asymptotic limit. We observe that the discrepancy is already small for vocabulary sizes on the order of $|V| \approx 500$, and it decreases as $|V|$ continues to increase. This suggests that, for the vocabulary sizes encountered in contemporary language models, the asymptotic formula offers an accurate approximation.

*Table 1.* Numerical values of the hypergeometric function and errors

| $\gamma$ | $\delta$ | $|V|$ | $_2F_1$ | Limit | Error |
|---|---|---|---|---|---|
| 0.200000 | 0.500000 | 500 | 0.884438 | 0.885156 | 0.000718 |
| 0.200000 | 1.000000 | 500 | 0.743109 | 0.744238 | 0.001129 |
| 0.200000 | 2.000000 | 500 | 0.438154 | 0.439018 | 0.000864 |
| 0.300000 | 0.500000 | 500 | 0.836557 | 0.837089 | 0.000531 |
| 0.300000 | 1.000000 | 500 | 0.659165 | 0.659855 | 0.000690 |
| 0.300000 | 2.000000 | 500 | 0.342491 | 0.342851 | 0.000361 |
| 0.500000 | 0.500000 | 500 | 0.754803 | 0.755081 | 0.000279 |
| 0.500000 | 1.000000 | 500 | 0.537616 | 0.537883 | 0.000267 |
| 0.500000 | 2.000000 | 500 | 0.238319 | 0.238406 | 0.000087 |
| 0.700000 | 0.500000 | 500 | 0.687582 | 0.687708 | 0.000126 |
| 0.700000 | 1.000000 | 500 | 0.453872 | 0.453968 | 0.000096 |
| 0.700000 | 2.000000 | 500 | 0.182714 | 0.182737 | 0.000023 |
| 0.200000 | 0.500000 | 1000 | 0.884797 | 0.885156 | 0.000359 |
| 0.200000 | 1.000000 | 1000 | 0.743672 | 0.744238 | 0.000566 |
| 0.200000 | 2.000000 | 1000 | 0.438586 | 0.439018 | 0.000432 |
| 0.300000 | 0.500000 | 1000 | 0.836823 | 0.837089 | 0.000266 |
| 0.300000 | 1.000000 | 1000 | 0.659510 | 0.659855 | 0.000345 |
| 0.300000 | 2.000000 | 1000 | 0.342671 | 0.342851 | 0.000180 |
| 0.500000 | 0.500000 | 1000 | 0.754942 | 0.755081 | 0.000140 |
| 0.500000 | 1.000000 | 1000 | 0.537749 | 0.537883 | 0.000134 |
| 0.500000 | 2.000000 | 1000 | 0.238363 | 0.238406 | 0.000043 |
| 0.700000 | 0.500000 | 1000 | 0.687645 | 0.687708 | 0.000063 |
| 0.700000 | 1.000000 | 1000 | 0.453920 | 0.453968 | 0.000048 |
| 0.700000 | 2.000000 | 1000 | 0.182726 | 0.182737 | 0.000012 |

### A.5. Proof of Lemma 3.9

**Plug-In Interpretation via Bernoulli KL Divergence**   Under the plug-in approximation $P(x \in G) \approx \gamma$, consider the result from the perspective of the binary statistic

$$I = \mathbf{1}_{x \in G}.$$

$$I \sim \text{Bernoulli}(\gamma) \quad \text{under } \mathbb{P},$$

$$I \sim \text{Bernoulli}(\gamma') \quad \text{under } \mathbb{P}_\delta.$$

The KL divergence between these two Bernoulli distributions is straightforward:

$$\text{KL}\big(\text{Bern}(\gamma')\|\text{Bern}(\gamma)\big) = \gamma' \log \frac{\gamma'}{\gamma} + (1 - \gamma') \log \frac{1 - \gamma'}{1 - \gamma}.$$

From

$$\gamma' = \frac{\gamma e^\delta}{1 + \gamma(e^\delta - 1)},$$

substituting this expression into the previous formula yields the stated conclusion.

**Proof of KL Monotonicity for $\delta$**   Recall

$$D_{\text{KL}}(\gamma, \delta) = \delta \, \gamma'(\gamma, \delta) - \log\big(1 + \gamma(e^\delta - 1)\big), \qquad \gamma'(\gamma, \delta) = \frac{e^\delta \gamma}{1 + \gamma(e^\delta - 1)}.$$

Write

$$D(\delta) := 1 + \gamma(e^\delta - 1), \qquad D'(\delta) = \gamma e^\delta.$$

First, differentiate $\gamma'$:

$$\frac{\partial \gamma'}{\partial \delta} = \frac{\gamma e^\delta D - \gamma e^\delta D'}{D^2} = \frac{\gamma e^\delta (D - \gamma e^\delta)}{D^2} = \frac{\gamma e^\delta (1 - \gamma)}{D^2}.$$

Next note that

$$\frac{\partial}{\partial \delta} \log D = \frac{D'}{D} = \frac{\gamma e^\delta}{D} = \gamma'.$$

Therefore, by the product rule,

$$\frac{\partial}{\partial \delta} D_{\mathrm{KL}}(\gamma, \delta) = \gamma' + \delta \frac{\partial \gamma'}{\partial \delta} - \frac{\partial}{\partial \delta} \log D$$

$$= \gamma' + \delta \frac{\partial \gamma'}{\partial \delta} - \gamma'$$

$$= \delta \frac{\partial \gamma'}{\partial \delta}.$$

Substituting the expression for $\partial \gamma' / \partial \delta$ gives

$$\frac{\partial}{\partial \delta} D_{\mathrm{KL}}(\gamma, \delta) = \delta \cdot \frac{\gamma e^\delta (1 - \gamma)}{D(\delta)^2}.$$

Since $\delta > 0$, $\gamma \in (0, 1)$, $e^\delta > 0$, and $D(\delta) > 0$, the right-hand side is strictly positive. Hence $D_{\mathrm{KL}}(\gamma, \delta)$ is strictly increasing in $\delta$ for all $\delta > 0$.

**Proof of KL Non-Monotonicity for $\gamma$**

$$D_{\mathrm{KL}}(\gamma, \delta) = \delta \, \gamma'(\gamma, \delta) - \log D, \qquad \gamma'(\gamma, \delta) = \frac{e^\delta \gamma}{1 + \gamma(e^\delta - 1)}, \qquad D = 1 + \gamma(e^\delta - 1).$$

Let $a := e^\delta - 1 > 0$. Then $D = 1 + a\gamma$ and

$$\gamma' = \frac{e^\delta \gamma}{1 + a\gamma}.$$

Differentiate with respect to $\gamma$:

$$\frac{\partial \gamma'}{\partial \gamma} = \frac{e^\delta (1 + a\gamma) - e^\delta \gamma a}{(1 + a\gamma)^2} = \frac{e^\delta}{(1 + a\gamma)^2}, \qquad \frac{\partial}{\partial \gamma} \log D = \frac{a}{1 + a\gamma}.$$

Hence

$$\frac{\partial}{\partial \gamma} D_{\mathrm{KL}}(\gamma, \delta) = \delta \frac{e^\delta}{(1 + a\gamma)^2} - \frac{a}{1 + a\gamma} = \frac{N(\gamma)}{(1 + a\gamma)^2},$$

where

$$N(\gamma) := \delta e^\delta - a(1 + a\gamma).$$

Observe that $N(\gamma)$ is a strictly decreasing linear function of $\gamma$ since $N'(\gamma) = -a^2 < 0$. Denote

$$\gamma_0(\delta) = \frac{\delta e^\delta - (e^\delta - 1)}{(e^\delta - 1)^2}$$

the solution for $N(\gamma_0) = 0$. Since the denominator $(1 + a\gamma)^2 > 0$,

$$\frac{\partial}{\partial \gamma} D_{\mathrm{KL}}(\gamma, \delta) > 0 \quad (\gamma < \gamma_0), \qquad \frac{\partial}{\partial \gamma} D_{\mathrm{KL}}(\gamma, \delta) < 0 \quad (\gamma > \gamma_0).$$

Therefore $D_{\mathrm{KL}}(\gamma, \delta)$ first increases and then decreases in $\gamma$, and hence is not monotonic in $\gamma$.

### A.6. Derivation of $\gamma^*$

Recall the power formula Eq. 1

$$\pi^*(\gamma, \delta) \approx \Phi\left(\frac{\sqrt{n}(\gamma' - \gamma) - z_{1-\alpha}\sqrt{\gamma(1-\gamma)}}{\sqrt{c\gamma'(1-\gamma')}}\right)$$

To achieve a power greater than 0.5, one would require the input to be positive, that is

$$\sqrt{n}(\gamma' - \gamma) - z_{1-\alpha}\sqrt{\gamma(1-\gamma)} > 0$$

Relaxing $\gamma'$ to 1,

$$\sqrt{n}(1 - \gamma) - z_{1-\alpha}\sqrt{\gamma(1-\gamma)} > 0$$

Therefore,

$$\gamma < \gamma^* = \frac{n}{n + z_{1-\alpha}^2}$$

## B. Additional Experiments and Experiment Settings

### B.1. Baseline Methods

Table 2 summarizes the baseline watermarking methods evaluated in our experiments.

*Table 2.* Baseline watermarking methods used for comparison.

| Method | Reference | Description |
|---|---|---|
| DP | (Cai et al., 2024) | KL divergence - Difference in probability optimization |
| OPT-style | (Wouters, 2024) | Log perplexity - Difference in green-token optimization |
| Grid Search | – | Exhaustive heuristic search over watermark parameter grid |
| Ours | This work | KL divergence - Statistical power optimization |

### B.2. Semantic Measurements

**BLEU** (Papineni et al., 2002) is a precision-oriented metric that computes the overlap of n-grams between the generated text and reference text. It is widely used to measure generation accuracy and fluency. BLEU scores are computed using the NLTK implementation, which follows the original formulation by (Papineni et al., 2002).

**ROUGE** (Lin, 2004) is recall-oriented and measures how much of the reference content is covered by the generated text, making it particularly suitable for evaluating content preservation and coverage. ROUGE scores are calculated using the Python `rouge-score` library, a standard reimplementation of the ROUGE metric.

**BERTScore** (Zhang et al., 2020) is an evaluation metric that leverages contextual embeddings from pre-trained bidirectional Transformer encoders (BERT) (Devlin et al., 2019) to measure semantic similarity between candidate and reference texts. Unlike BLEU or ROUGE, which rely on surface-level n-gram overlap, BERTScore captures deeper semantic correspondence through embedding-based matching. BERTScore is computed using the official `bert-score` package.

### B.3. Hyperparameter Ranges

Table 3 reports the hyperparameter ranges used for all baseline methods and our proposed approach.

*Table 3.* Hyperparameter search ranges for all methods.

| Method | Parameter | Symbol | Values |
|--------|-----------|--------|--------|
| DP | Distortion budget | $\Delta$ | $\{0.1, 0.2, 0.3, 0.4, 0.5, 0.6, 0.7, 0.8, 0.9\}$ |
| OPT-style | Green-list ratio | $\gamma$ | $\{0.1, 0.2, 0.3, 0.4, 0.5, 0.6, 0.7, 0.8, 0.9\}$ |
| | Logit bias scale | $\beta$ | $\{0.5, 1, 2, 5, 10\}$ |
| Grid Search | Green-list ratio | $\gamma$ | $\{0.1, 0.2, 0.3, 0.4, 0.5, 0.6, 0.7, 0.8, 0.9\}$ |
| | Logit bias strength | $\delta$ | $\{0.5, 1, 2, 5, 10\}$ |
| Ours | Search initializer | $\gamma^*$ | $n/(n + z_{1-\alpha}^2)$ |
| | KL Budget | $D_{\mathrm{KL}}$ | $-\log \gamma^* + 0.025$ |
| | Reduced-strength tradeoff | $(\gamma, \delta)$ | $\gamma \in \{0.1, 0.2\}, \delta \in \{1, 2, 3, 4, 5\}$ |

## B.4. Optimization Details

For our method, the constrained optimization problem is solved numerically under a KL budget constraint. In addition to the optimal solution, we evaluate reduced-strength configurations to visualize the distortion-detectability trade-off curve. For the closed-form power-calibration configuration, we set $c = 1$; empirical estimation of $c$ is discussed in Appendix C.

## B.5. Implementation Consistency and Fairness

All methods are implemented using a unified generation and detection pipeline based on the public KGW implementation (Kirchenbauer et al., 2024b).

For all baselines except OPT and our method, we use the same sampling strategy, detection statistic, and evaluation protocol. For OPT, the only difference is in the watermark generation phase, where we implemented their original approach; all other components remain the same. This design ensures that observed performance differences are attributable to watermark parameterization rather than implementation artifacts. For the baselines, we explored a broad and comprehensive range of settings beyond those used in prior work to ensure optimal performance.

We use the n-gram filtering from (Fernandez et al., 2023) for better result accuracy.

## B.6. Z-Score Detector and Significance Level

We use the standard KGW detection statistic based on the normalized green-token count. Under $H_0$, the indicator variables form an i.i.d. Bernoulli($\gamma$) process, and the resulting sum admits a normal approximation, which motivates the z-score formulation. Given a generated sequence of length $n$, the test statistic is defined as

$$S_n = \sum_{i=1}^{n} \mathbf{1}\{x_i \in G_i\},$$

which is standardized as

$$Z = \frac{S_n - n\gamma}{\sqrt{n\gamma(1-\gamma)}}.$$

Detection is performed by thresholding $Z$ at $z_{1-\alpha}$, the $(1-\alpha)$ quantile of the standard normal distribution.

We set $\alpha = 0.05$, which is a standard choice in hypothesis testing and is consistent with prior watermarking evaluations.

**Short-Sequence Evaluation.** Prior work shows that KGW-style watermark statistics perform well with increasing sequence length due to signal accumulation (Kirchenbauer et al., 2024b), leading to artificially inflated detectability. To avoid this effect and better isolate statistical efficiency differences between parameter selection methods, we evaluate detection on short generations with $n = 50$ tokens.

**Long-Form Generation and the Dual Formulation.** For long-form generation, the detection signal accumulates with sequence length: in the power approximation, the watermark-induced separation is amplified by the $\sqrt{n}$ term. This motivates

the dual formulation, which fixes a target detection power and searches for the smallest per-token KL budget $K_0$. Long sequences can therefore maintain detectability through signal accumulation while reducing per-token distortion, rather than relying on a stronger watermark at each token.

### B.7. Detector Modularity and Signal-Removal Robustness

Although the closed-form calibration in the main text is instantiated with the standard full-sequence $z$-score detector, the main distributional input is the watermark-induced signal itself: the shifted green-token probability $\gamma'(\gamma, \delta)$ and the sequence-level behavior of the indicator process $\{I_i\}$. Therefore, the same calibration framework can be paired with other KGW detectors by replacing the aggregation rule and rejection threshold. For a detector $D$, let $A_D(I_{1:n})$ denote its aggregation rule applied to the green-token indicator sequence and let $z_{1-\alpha}^D$ denote its calibrated null threshold. Once $\gamma'(\gamma, \delta)$ and the long-run variance behavior of $I_{1:n}$ are available, the detector-specific power can be written as

$$\pi_D(\gamma, \delta) = \mathbb{P}_\delta \{A_D(I_{1:n}) > z_{1-\alpha}^D\}.$$

The full-sequence $z$-score admits the closed-form approximation in Eq. 1. Other KGW detectors can be handled by changing $A_D$ and calibrating $z_{1-\alpha}^D$ under the null, while preserving the same underlying signal model.

This modular view is useful for common post-processing attacks. Paraphrasing, copy-paste insertion, deletion, and local rewriting differ in surface form, but from the keyed detector's perspective they can reduce, dilute, or remove the green-list bias in the affected region. We model such attacks through a signal-removal abstraction: a contiguous fraction $f$ of the generated text is replaced by unwatermarked text, leaving only $(1-f)n$ positions with the watermark-induced green-token bias.

For localized attacks, we consider the complement of the WinMax detector (Kirchenbauer et al., 2024b). Instead of selecting the best contiguous block to keep, WinMax-C selects the best contiguous block $\mathcal{C}$ to remove and scores the remaining tokens:

$$Z_{\text{WMC}} = \max_{\mathcal{C}} \frac{\sum_{i \notin \mathcal{C}} I_i - \gamma(n - |\mathcal{C}|)}{\sqrt{(n - |\mathcal{C}|)\gamma(1 - \gamma)}}.$$

This is useful when an interior segment has been attacked, because the surviving watermark signal may lie on both sides of the attacked segment.

The $\sqrt{1-f}$ scaling follows directly from the normalization. In the clean case, the expected full-sequence signal shift is

$$\Delta_{\text{clean}} = \frac{n(\gamma' - \gamma)}{\sqrt{n\gamma(1 - \gamma)}} = \frac{\sqrt{n}(\gamma' - \gamma)}{\sqrt{\gamma(1 - \gamma)}}.$$

After signal removal, the full-sequence $z$-score still normalizes by $\sqrt{n\gamma(1 - \gamma)}$, but only $(1-f)n$ tokens carry watermark signal. Hence

$$\Delta_{\text{full}} = \frac{(1 - f)n(\gamma' - \gamma)}{\sqrt{n\gamma(1 - \gamma)}} = (1 - f)\Delta_{\text{clean}}.$$

By contrast, WinMax-C removes the attacked block and scores only the retained $m = (1 - f)n$ tokens. Its normalization is therefore $\sqrt{m\gamma(1 - \gamma)}$, giving

$$\Delta_{\text{WMC}} = \frac{m(\gamma' - \gamma)}{\sqrt{m\gamma(1 - \gamma)}} = \sqrt{1 - f}\,\Delta_{\text{clean}}.$$

Thus WinMax-C retains a signal scaling of $\sqrt{1-f}$, while the standard full-sequence $z$-score retains approximately $1 - f$. Equivalently, WinMax-C has a $1/\sqrt{1-f}$ advantage in expected signal over the full-sequence $z$-score under this localized signal-removal model.

*Table 4.* Largest $-\log(D_{\mathrm{KL}})$ at which each method maintains at least 0.95 post-attack TPR using WinMax-C on C4, among configurations with clean TPR at least 0.95. Larger values correspond to lower KL distortion.

| Attack | Model | DP | GRID | OPT | Ours |
|--------|-------|------|------|------|------|
| 10% | GPT-2 | 1.97 | 2.15 | 2.03 | 2.70 |
| 10% | OPT-125M | 1.39 | 1.86 | 1.70 | 2.53 |
| 10% | Pythia-160M | 1.34 | 2.22 | 2.13 | 2.70 |
| 15% | GPT-2 | 1.97 | 2.16 | 2.01 | 2.69 |
| 15% | OPT-125M | 1.38 | 1.86 | 1.65 | 2.56 |
| 15% | Pythia-160M | 1.34 | 2.22 | 2.12 | 2.70 |
| 20% | GPT-2 | 1.98 | 2.16 | 2.01 | 2.69 |
| 20% | OPT-125M | 1.39 | 1.87 | 1.68 | 2.56 |
| 20% | Pythia-160M | 1.34 | 2.22 | 2.13 | 2.68 |

Across attack levels and model families, the calibrated parameters preserve high post-attack TPR at lower distortion than the baselines. These results support using the clean-generation calibration as the default parameter choice and switching to detector-specific aggregation, such as WinMax-C, when localized editing robustness is required.

### B.8. KL Divergence Computation

At each generation step $i$, let $\mathbb{P}_i$ and $\mathbb{P}_{\delta,i}$ denote the unwatermarked and watermarked next-token distributions obtained from the model logits via softmax. We compute the token-wise KL divergence

$$D_{\mathrm{KL}}(\mathbb{P}_{\delta,i} \,\|\, \mathbb{P}_i) = \sum_{v \in V} \mathbb{P}_{\delta,i}(v) \log \frac{\mathbb{P}_{\delta,i}(v)}{\mathbb{P}_i(v)}.$$

The reported distortion score is obtained by averaging this quantity over all generated positions:

$$D_{\mathrm{KL}} = \frac{1}{n} \sum_{i=1}^{n} D_{\mathrm{KL}}(\mathbb{P}_{\delta,i} \,\|\, \mathbb{P}_i).$$

**Relation to Theoretical Analysis.** Our theoretical analysis derives a closed-form expression for the KL divergence between the induced Bernoulli distributions of the green-token indicator under watermarking and non-watermarking. This quantity captures the distortion of the binary detection statistic and serves as a low-dimensional surrogate for watermark strength.

In experiments, we report the full-vocabulary KL divergence between next-token distributions, which directly measures distributional shift at the model output level.

**Implementation Details.** KL divergence is computed directly from the model logits before sampling, without truncation of the vocabulary distribution.

### B.9. $R^2$ of Theoretical Values and Empirical Values

To assess the consistency between the theoretical analysis and empirical observations, we examine the linear relationship between theoretical predictions and measured empirical values. The theory implies a strict proportional relationship between the two quantities, i.e., when the theoretical value is zero, the empirical value should also be zero. Therefore, we perform *linear regression without an intercept*, using the model $y = kx$ rather than the general form $y = kx + b$. Under this formulation, the coefficient of determination $R^2$ directly reflects how well the theoretical quantity explains the variation in the empirical measurements, both in trend and magnitude.

Table 5 reports the resulting $R^2$ values across different models (OPT-125M, Pythia-160M, GPT-2) and datasets (C4, LFQA, Wikipedia) for both the *green-token rate* metric and the *KL divergence* metric. Data are collected with parameters $\gamma \in \{0.1, 0.2, 0.3, 0.4, 0.5, 0.6, 0.7, 0.8, 0.9\}$ and $\delta \in \{0.5, 1, 2\}$. In all settings, the $R^2$ values are close to 1 (mostly above 0.99), indicating that the empirical results can be well characterized as a linear scaling of the theoretical predictions.

This demonstrates that the derived theoretical relationship is both qualitatively and quantitatively precise. Moreover, the consistency of these high $R^2$ values across model architectures and data distributions suggests that the relationship captures a general underlying mechanism rather than a dataset- or model-specific one.

*Table 5.* $R^2$ for green-token rate and KL metrics

| Model | Dataset | Green-token rate $R^2$ | KL $R^2$ |
|---|---|---|---|
| opt-125m | c4 | 0.9967 | 0.9980 |
| opt-125m | lfqa | 0.9949 | 0.9965 |
| opt-125m | wikipedia | 0.9959 | 0.9947 |
| pythia-160m | c4 | 0.9892 | 0.9939 |
| pythia-160m | lfqa | 0.9881 | 0.9951 |
| pythia-160m | wikipedia | 0.9962 | 0.9943 |
| gpt2 | c4 | 0.9933 | 0.9984 |
| gpt2 | lfqa | 0.9920 | 0.9958 |
| gpt2 | wikipedia | 0.9949 | 0.9957 |

## B.10. Distributional Verification

To assess the goodness-of-fit of the normal approximation, we construct Q–Q plots comparing the empirical quantiles of the standardized green-token count statistic with the theoretical quantiles of the standard normal distribution.

Figure 3 shows results for sequences generated with $n = 500$ tokens and green-list ratio $\gamma = 0.2$. The left panel corresponds to the null hypothesis ($\delta = 0$), and the right panel corresponds to the alternative hypothesis ($\delta = 1$).

In both cases, the empirical quantiles closely follow the identity line, indicating strong agreement with the normal approximation. Minor deviations in the tails are expected for finite sample sizes and do not materially affect detection.

These results provide empirical support for the Gaussian approximation in our detection framework.

## B.11. Alternative Calibration

To validate the theoretical characterization of watermark-induced green-token probability shifts, we conduct experiments across a wide range of watermark parameters with $\gamma \in \{0.1, 0.2, 0.3, 0.4, 0.5, 0.6, 0.7, 0.8, 0.9\}$ and $\delta \in \{0.5, 1, 2\}$.

For each configuration, we compute the empirical green-token rate and compare it to the corresponding theoretical value derived. We fit a linear regression model without an intercept, $y = kx$, reflecting the theoretical prediction that the relationship is proportional.

Figure 4 shows the comparison across the C4, LFQA, and Wikipedia datasets. The results exhibit strong linear alignment between empirical measurements and theoretical predictions.

To further quantify this agreement, we report the coefficient of determination ($R^2$) values for both green-token rate and KL divergence across all models and datasets in Table 5. In all settings, $R^2$ exceeds 0.98, indicating that the theoretical analysis accurately captures the empirical behavior across architectures and data distributions.

## B.12. Pareto Frontier Construction

For visualization, Pareto frontier lines are constructed using Pareto points to fit the curve

$$y = 1/(1 + e^{ax-b})$$

The curve starts near $(-\infty, 1)$ because sufficiently large separation yields high power. A smooth monotone curve is fitted to the frontier for presentation purposes only and does not affect the reported evaluation results.

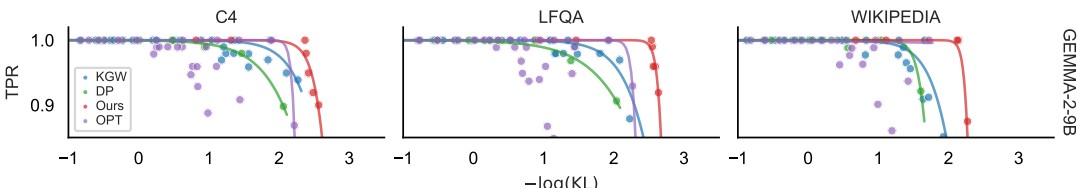

*Figure 7.* Statistical power (measured by TPR) versus semantic distortion (measured by KL divergence) on Gemma-2 9B. Results are obtained on all three datasets. For each method, the curve is fitted using its Pareto frontier with a small tolerance allowed for visualization quality. The fitted curve follows the form $y = 1/(1 + e^{ax - b})$.

**Additional Large-Model Validation.** Figure 7 reports the corresponding Pareto frontier results for Gemma-2 9B. The larger-model setting exhibits the same qualitative detectability-distortion pattern as the main-model results in Figure 6.

### B.13. Text Quality Metrics

BLEU scores are computed using the NLTK implementation. ROUGE scores are obtained using the `rouge-score` library. BERTScore is computed using the official `bert-score` package with pretrained BERT encoders.

All metrics are computed between watermarked and unwatermarked outputs generated from identical prompts.

To quantify how the KL distortion proxy relates to sequence-level quality, we fit linear regressions of BLEU, ROUGE, and BERTScore against empirical KL divergence using all experimental configurations. The resulting coefficients of determination are $R^2 = 0.566$ for BLEU, $R^2 = 0.870$ for ROUGE, and $R^2 = 0.791$ for BERTScore. These results indicate that KL is a useful distributional proxy for sequence-level quality, while also reflecting that surface-form metrics such as BLEU capture additional variation beyond token-level KL.

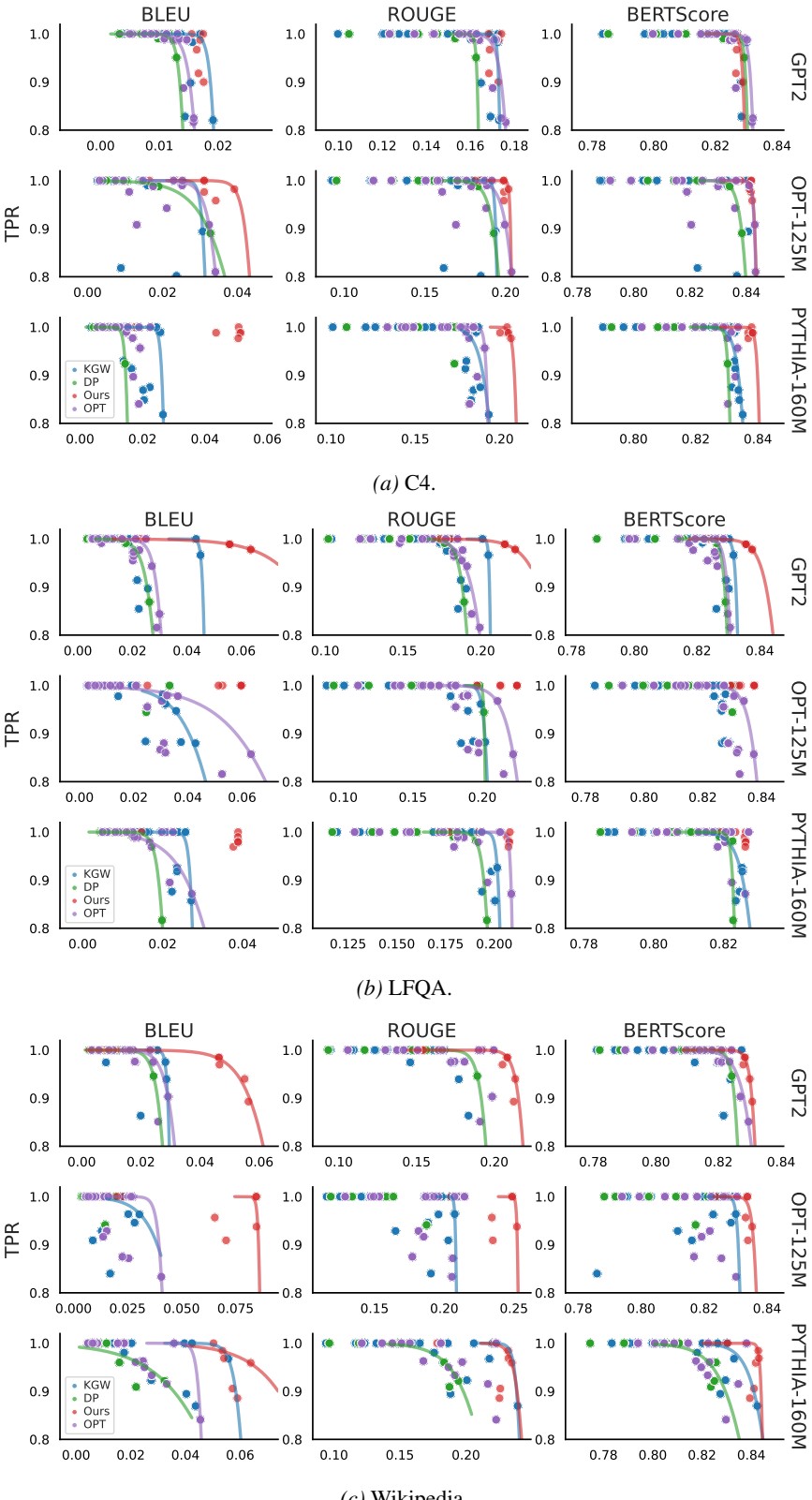

*(a)* C4.

*(b)* LFQA.

*(c)* Wikipedia.

*Figure 8.* Complete statistical power (TPR) versus semantic quality results under BLEU, ROUGE, and BERTScore. Rows correspond to models and columns correspond to quality metrics. Higher values indicate better quality for all metrics. For each method, the curve is fitted using its Pareto frontier with a small tolerance allowed for visualization quality.

# C. Estimation of the Variance Inflation Constant $c$

## C.1. Background

In practice, tokens generated by language models are *not independent* due to contextual and semantic dependencies. To account for deviation from the i.i.d. assumption, we introduce a *variance inflation constant $c$*:

$$\mathrm{Var}(S_n) = c \cdot n\, \gamma'(1 - \gamma').$$

The interpretation of $c$ is:

- $c = 1$: IID behavior (no correlation),

- $c > 1$: positive correlation between tokens (variance inflated),

- $c < 1$: negative correlation (variance deflated).

## C.2. Estimation Procedure

Suppose we collect $M$ independent sequences generated under the same watermark configuration $(\gamma, \delta, n)$. For each sequence $j$, define the total number of green tokens as

$$S_j = \sum_{t=1}^{n} I_{j,t}, \quad j = 1, \ldots, M,$$

where $I_{j,t}$ is the indicator variable denoting whether the $t$-th token in sequence $j$ is classified as green.

The empirical mean green-token count is

$$\bar{S} = \frac{1}{M} \sum_{j=1}^{M} S_j,$$

and the unbiased sample variance across sequences is

$$\widehat{\mathrm{Var}}(S) = \frac{1}{M-1} \sum_{j=1}^{M} (S_j - \bar{S})^2.$$

From the sample mean count, we obtain an empirical estimate of the effective green-token rate:

$$\hat{\gamma}' = \frac{\bar{S}}{n}.$$

To quantify the deviation from the ideal i.i.d. Bernoulli assumption, we estimate a *variance inflation constant*

$$\hat{c} = \frac{\widehat{\mathrm{Var}}(S)}{n\, \hat{\gamma}'(1 - \hat{\gamma}')}.$$

Under the i.i.d. model, the denominator $n\, \hat{\gamma}'(1 - \hat{\gamma}')$ represents the theoretical variance of a binomial random variable with parameters $n$ and $\hat{\gamma}'$. Therefore, $\hat{c} \approx 1$ indicates consistency with independence, while $\hat{c} > 1$ reveals overdispersion induced by token dependence and model dynamics. This statistic thus provides a practical uncertainty quantification tool for assessing violations of the i.i.d. assumption in real model outputs.

We conducted experiments with $\gamma = 0.2$ and $\delta = 1$ to empirically validate this variance inflation phenomenon. Beyond confirming systematic overdispersion, we further observed *distinct model- and dataset-dependent patterns* in $\hat{c}$.

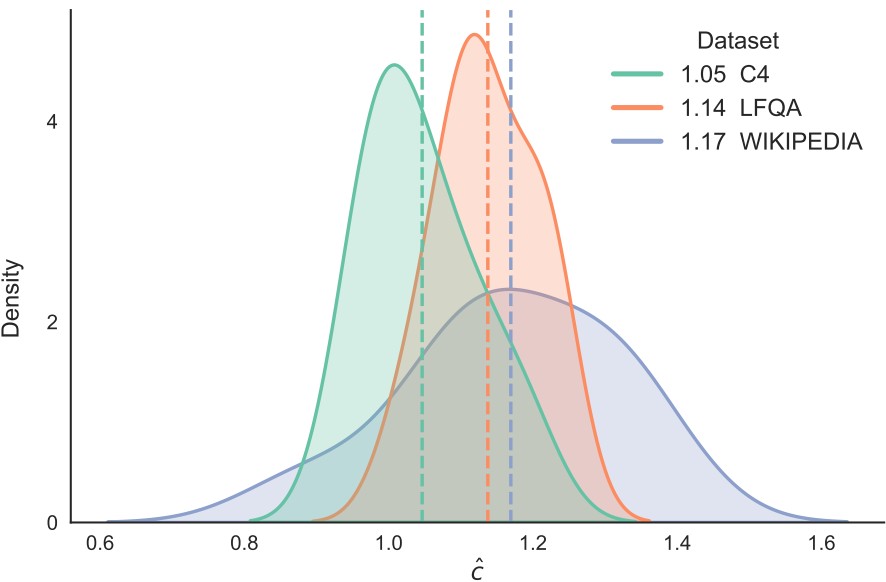

*Figure 9.* Smoothed inflation constant distribution, categorized by dataset

The variance inflation constant $c$ can be interpreted as a signature of dependence induced by the underlying generation mechanism, and therefore naturally varies across datasets. Among the datasets considered, Wikipedia yields the largest observed inflation. Wikipedia articles are typically topic-stable, stylistically regular, and logically coherent over long spans. These properties reduce the effective token choice space at each step and create persistent semantic trajectories. Consequently, the cumulative covariance in $\mathrm{Var}(S)$ is increased, explaining why

$$c_{\text{Wikipedia}} > c_{\text{LFQA}} > c_{\text{C4}}.$$

This demonstrates that variance inflation reflects how strongly the data distribution enforces semantic continuity.

## D. Limitations

While our framework demonstrates strong performance, our theoretical assumptions warrant further discussion. First, although the information decay assumption facilitates asymptotic analysis, the memory mechanisms in modern Transformers may result in a slow-decay scenario. This suggests that in non-asymptotic regimes, the theoretical convergence could be less precise than idealized models predict. Second, in low-entropy domains such as code generation, where next-token probabilities approach singularity, the inherent randomness required for hiding the watermark is naturally constrained, suggesting a boundary where the stationarity assumption may need adaptation. Finally, our optimization targets KL divergence, a distributional metric. Because semantic distances between tokens are non-uniform, there remains room for semantic-aware watermarking designs.

Watermarking systems also introduce deployment risks beyond statistical calibration. False positives can create false accusations if detector outputs are treated as definitive evidence, especially in high-stakes settings such as education, hiring, or content moderation. Conversely, adversaries may attempt paraphrasing, editing, or signal-removal attacks to reduce detectability; accordingly, watermark evidence should not be interpreted as a complete provenance guarantee. Practical deployments should therefore pair calibrated false-positive control with transparent reporting of detector uncertainty, clear scope limitations, and human review when decisions materially affect users.

## E. Demonstration of $\gamma_0$

We show that using $\gamma < \gamma_0(\delta)$ does not yield optimal results. Red dots in the figures represent lower values of $\gamma$, and the results align with our theory.

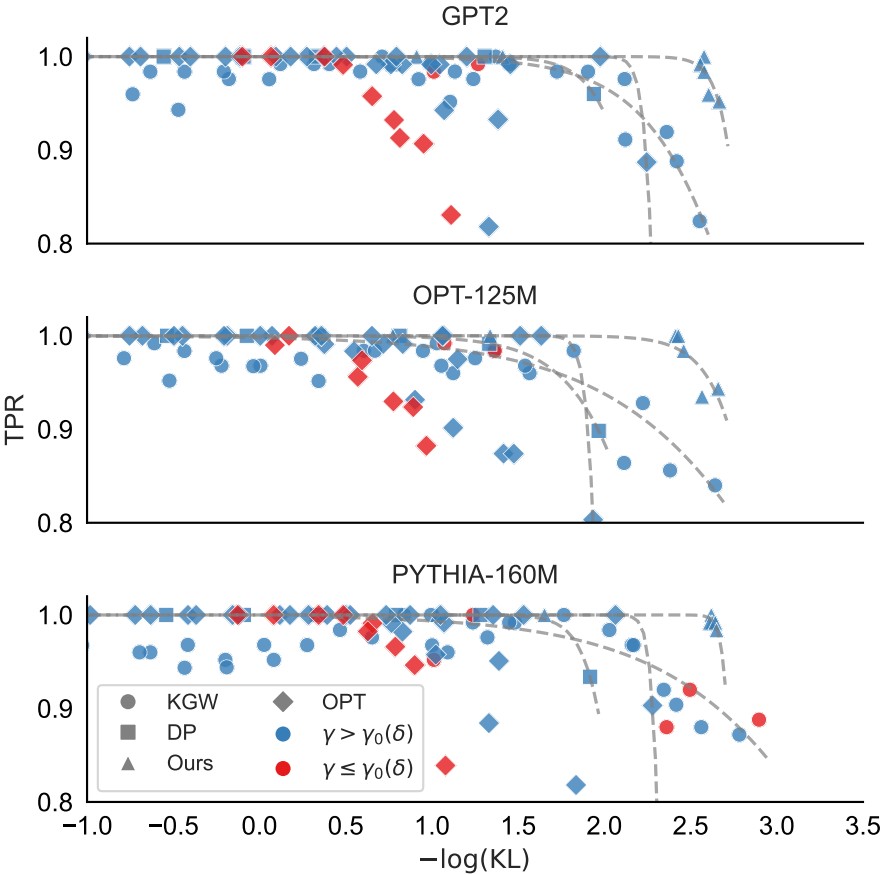

*Figure 10.* Statistical power (measured by TPR) versus semantic distortion (measured by KL divergence). Results are obtained on the C4 dataset. For each method, the curve is fitted using its Pareto frontier with a small tolerance allowed for visualization quality. The fitted curve follows the form $y = 1/(1 + e^{ax-b})$. This figure is colored according to the relative values of $\gamma$ and $\gamma_0(\delta)$.

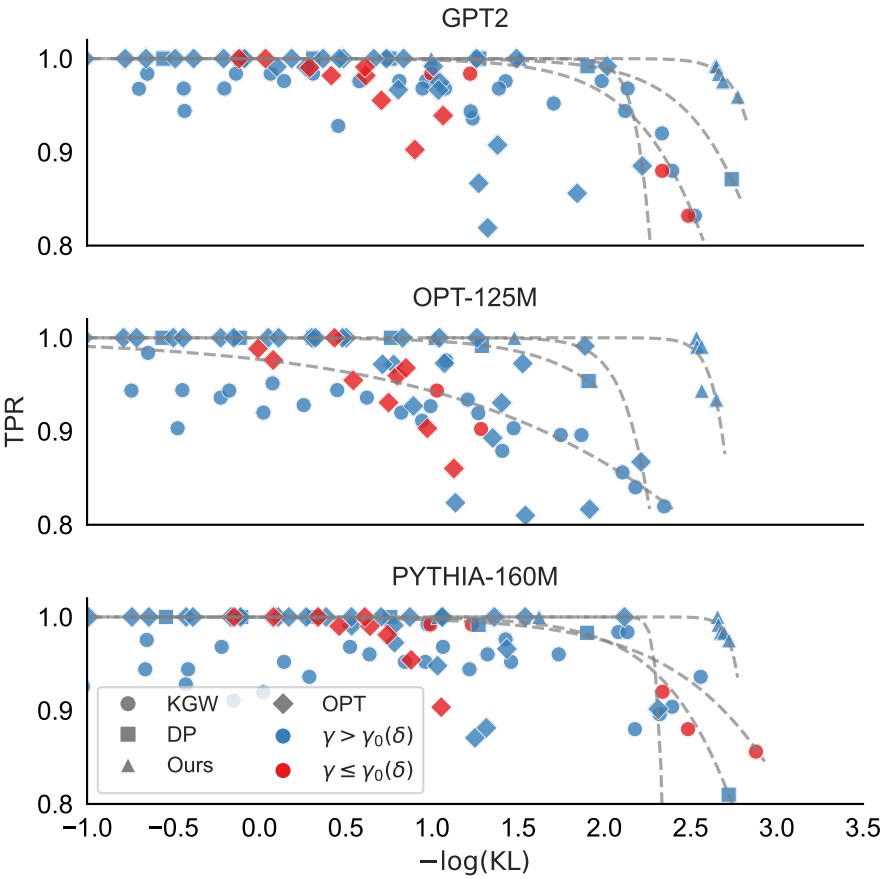

*Figure 11.* Statistical power (measured by TPR) versus semantic distortion (measured by KL divergence). Results are obtained on the LFQA dataset. For each method, the curve is fitted using its Pareto frontier with a small tolerance allowed for visualization quality. The fitted curve follows the form $y = 1/(1 + e^{ax-b})$. This figure is colored according to the relative values of $\gamma$ and $\gamma_0(\delta)$.

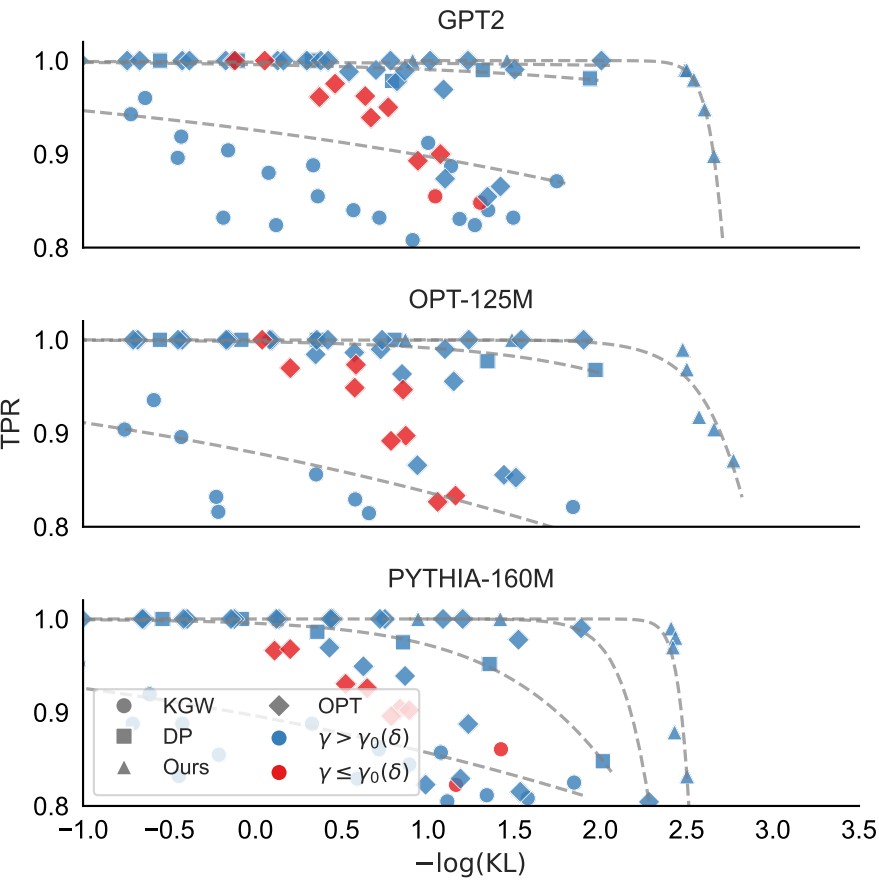

*Figure 12.* Statistical power (measured by TPR) versus semantic distortion (measured by KL divergence). Results are obtained on the Wikipedia dataset. For each method, the curve is fitted using its Pareto frontier with a small tolerance allowed for visualization quality. The fitted curve follows the form $y = 1/(1 + e^{ax-b})$. This figure is colored according to the relative values of $\gamma$ and $\gamma_0(\delta)$.

