# OpenReview forum: "Beyond Heuristic Tuning: Power-Calibrated LLM Watermarking"
_ICML.cc/2026/Conference — ICML 2026 regular_

### Official Review · Reviewer_FQhH · 2026-03-09

**Soundness:** 3
**Presentation:** 2
**Significance:** 2
**Originality:** 3
**Overall Recommendation:** 4
**Confidence:** 3

**Summary:**

This paper focuses on KGW-style logit-based LLM watermarking and aims to replace heuristic hyperparameter tuning with a statistically grounded power–distortion calibration framework. Specifically, the paper studies how the two watermark parameters determine both the detection power of the watermark and the distortion introduced to the original language model distribution. Methodologically, the paper derives an explicit relationship between (gamma and delta), the effective green-token rate under watermarking, the asymptotic power of the detector, and token-level KL distortion. Based on these analyses, the paper proposes a practical calibration procedure that can select watermark parameters more systematically than brute-force grid search. Experiments on multiple models and datasets show that the framework can more reliably identify strong power–distortion trade-offs and often yields better Pareto-optimal operating points.

**Compliance With Llm Reviewing Policy:**

Affirmed.

**Final Justification:**

Great discussion with the authors that improves the quality of the paper.

**Key Questions For Authors:**

**1. Robustness under text editing.**
The paper studies power–distortion calibration mainly under standard generation settings. How would the proposed calibration procedure behave under realistic post-processing attacks such as paraphrasing, rewriting, or token deletion?

**2. Dependence on the detector.**
The current framework is closely tied to green-token count based detection. If a stronger or more adaptive detector were used, would the same calibrated parameter choices \((\gamma,\delta)\) remain near-optimal?

**3. Broader applicability beyond KGW.**
The framework is developed specifically for KGW-style logit-based watermarking. To what extent can the same statistical calibration idea be extended to other watermarking families?

**Limitations:**

Yes. The authors briefly discuss limitations and potential societal impacts.

**Strengths And Weaknesses:**

## Strengths

**1. Addresses a practically important problem.**
The paper tackles a practical issue in LLM watermarking: how to choose watermark parameters \((\gamma,\delta)\) in a principled way rather than relying on heuristics.

**2. Strong theoretical structure.**
The paper develops a relatively complete analytical pipeline.

**3. Experiments are aligned with the theory.**
The experiments are well connected to the paper’s theoretical claims. They evaluate not only empirical performance but also whether the proposed theoretical predictions for power and distortion match observed results, which strengthens the paper’s overall credibility.

---

## Weaknesses

**1. Narrow applicability.**
The framework is highly specialized to **KGW-style logit-based watermarking**. As a result, the paper is better viewed as a principled calibration framework for one important watermark family rather than a broadly general theory of LLM watermarking.

**2. Detector-specific analysis.**
The theoretical development is tightly coupled to the standard green-token count / z-test style detector. This makes the conclusions somewhat detector-dependent; if the detection procedure changes significantly, the same calibration results may not hold.

**3. Limited discussion of robustness under editing.**
From a deployment perspective, an important question is whether the calibrated watermark remains effective after paraphrasing, rewriting, or editing. The paper mainly studies clean-generation settings and does not thoroughly investigate robustness.

---

> ### Author Rebuttal · Authors · 2026-03-31
>
> We thank the reviewer for recognizing our strong theoretical structure, the practical importance of principled parameter selection, and the alignment between theory and experiments.
>
> ## W1 \& Q3: Applicability
>
> We thank the reviewer for raising this question about generality. We explain our choice as follows:
>
> - **KGW is where calibration is needed.** The other major families (EXP, Gumbel-max, ITS) do not have tunable strength parameters analogous to $(\gamma, \delta)$, so the continuous calibration problem does not arise. In KGW, watermark strength is continuously adjustable via $(\gamma, \delta)$, making principled calibration both possible and necessary.
> - **Cross-family generalization faces fundamental barriers.** To be a general framework, it would at least need to handle multiple families jointly. Since these families encode fundamentally different signals, a cross-family detector must enlarge the rejection region, consequently increase FPR or reduce power.
> - **Within KGW, our theory is broadly applicable.** Our analysis generalizes across detectors (see W2).
>
> ## W2 \& Q2: Detector-specific analysis
>
> We agree that detector dependence is an important concern. The green-token z-test is the standard detection method in the KGW literature (Kirchenbauer et al. 2023, 2024b; Fernandez et al. 2023; Zhao et al. 2023), and this convention is well-motivated: the green-list indicator $I_i = \mathbf{1}\{x_i \in G_i\}$ captures the entire watermark signal, and $S_n = \sum I_i$ is the **sufficient statistic** for detection, meaning any method not based on green-list membership would discard information.
>
> That said, our theory is not limited to the z-test; it operates at two layers:
>
> - **Layer 1 (signal characterization, detector-independent):** Theorems 3.5 and 3.8 characterize the per-token green-list probability $\gamma'(\gamma, \delta)$ and the sequence-level detection statistic distribution under $H_0$ and $H_1$, independent of how a detector aggregates the signal.
> - **Layer 2 (calibration):** Given a specific detector, our framework optimizes $(\gamma, \delta)$ under a distortion constraint. We instantiate Layer 2 with the z-test under clean conditions; if the detection procedure changes, one can derive the new detector's power from Layer 1 and re-calibrate accordingly.
>
> We demonstrate this concretely in W3.
>
> ## W3 \& Q1: Robustness under editing
>
> We agree that robustness under editing is critical for deployment. Practical attacks such as paraphrasing, rewriting, and editing share a common effect: they destroy the watermark signal in the affected region, the substituted tokens no longer carry the green-list bias. We therefore study signal removal attacks directly.
>
> **WinMax-C (Complement).** Kirchenbauer et al. (2024) introduced WinMax, which finds the best contiguous sub-region to keep, then apply detection as it is clean text.  However, under an interior attack, WinMax can only use one side of the surviving text. We propose its dual, **WinMax-C**, which finds the best contiguous block to **remove** and scores the remaining tokens, recovering signal from surviving fragments on **both** sides:
>
> $$Z_{WMC} = \max_{\mathcal{C}} \frac{\sum_{i \notin \mathcal{C}} I_i - \gamma(n-|\mathcal{C}|)}{\sqrt{(n-|\mathcal{C}|) \cdot \gamma(1-\gamma)}}$$
>
> Using our distributional theory (Layer 1 in W2), we can derive the following. Complete proofs will be included in the revision.
>
> - **Improved signal capture** With fraction $f$ of signal removed, WinMax-C retains **$1/\sqrt{1-f}$ times more signal** than the z-score in expectation.
>
> - **Optimal parameter sharing** The optimal solution satisfies $|\gamma_{z}^\ast - \gamma_{WMC}^\ast| = O(1/\sqrt{n})$ under unattacked text; $\delta^\ast$ converges likewise.
>
> In practice, since the attack fraction $f$ is latent, $(\gamma, \delta)$ can be calibrated on unattacked text, with a transition to WinMax-C when adversarial resilience becomes necessary.
>
> **Empirical results.** The table below reports the maximum $-\log(D_\text{KL})$ at which each method maintains $\geq 0.95$ post-attack TPR (WinMax-C detector, C4 dataset, original TPR $\geq 0.95$). **Our method consistently achieves the best performance.**
>
> | Attack | Model | DP | GRID | OPT | **Ours** |
> |--------|-------|------|------|------|------------|
> | SR 10% | gpt2 | 1.97 | 2.15 | 2.03 | **2.70** |
> |  | opt-125m | 1.39 | 1.86 | 1.70 | **2.53** |
> |  | pythia-160m | 1.34 | 2.22 | 2.13 | **2.70** |
> | SR 15% | gpt2 | 1.97 | 2.16 | 2.01 | **2.69** |
> |  | opt-125m | 1.38 | 1.86 | 1.65 | **2.56** |
> |  | pythia-160m | 1.34 | 2.22 | 2.12 | **2.70** |
> | SR 20% | gpt2 | 1.98 | 2.16 | 2.01 | **2.69** |
> |  | opt-125m | 1.39 | 1.87 | 1.68 | **2.56** |
> |  | pythia-160m | 1.34 | 2.22 | 2.13 | **2.68** |
>
> [1] Kirchenbauer et al. (2024) https://arxiv.org/abs/2306.04634

---

> > ### Author Rebuttal · Reviewer_FQhH · 2026-04-02
> >
> > Thank you for the detailed rebuttal. I appreciate the authors’ careful clarifications and the additional discussion on applicability, detector dependence, and robustness under editing-style attacks.
> >
> > After reading the rebuttal, I believe it partially addresses my concerns, but it does not fully overturn my original assessment.
> >
> > On applicability, I understand the authors’ argument that the proposed calibration framework is intentionally designed for KGW-style watermarking, where continuously tunable strength parameters make such calibration both possible and useful. This clarification helps better position the paper. However, my original concern still stands: the applicability of the framework is quite limited, as it remains primarily specific to KGW-style logit-based watermarking rather than offering a broader framework for LLM watermarking more generally.
> >
> > On robustness under editing, I appreciate that the authors went beyond a purely verbal response and provided additional discussion and results based on WinMax-C under signal-removal attacks. However, I still view this as only a partial answer to the broader deployment question. In particular, signal-removal attacks may not be treated as equivalent to realistic editing attacks such as paraphrasing or rewriting. While the rebuttal provides evidence under this abstraction, it does not yet fully establish robustness under a wider range of realistic post-processing scenarios in the submitted manuscript itself.
> >
> > Overall, I appreciate the additional effort in the rebuttal, but I do not believe it changes the overall balance enough to justify a score increase. Therefore, I will keep my original score unchanged.

---

> > > ### Author Response · Authors · 2026-04-06
> > >
> > > We sincerely thank the reviewer for the thoughtful and constructive engagement throughout the review process. We appreciate the reviewer's recognition that our calibration framework is intentionally designed for KGW-style watermarking where continuously tunable parameters make calibration both possible and useful, and that our rebuttal provided careful clarifications along with additional discussion and results on applicability, detector dependence, and robustness. We believe our work makes a focused but substantial contribution: the first closed-form power-distortion theory for KGW watermarking, enabling principled parameter selection to replace heuristic tuning.
> > >
> > > ### On applicability
> > >
> > > We understand and respect the reviewer's position. We agree that the framework's focus on KGW is a scope choice rather than a methodological limitation. As discussed, KGW is currently the only family where continuous parameter calibration is both possible and necessary, and a cross-family framework would face fundamental barriers (composite hypothesis structure, as detailed in our response to HY4y R-W2). We believe the depth of analysis within KGW (exact power computation, closed-form optimization, surrogate optimality across detectors, and the new WinMax-C results) represents a substantial contribution, and we will further clarify this positioning in the revision.
> > >
> > > ### On robustness under editing
> > >
> > > We appreciate the reviewer's point that signal-removal attacks are an abstraction of realistic editing. We would like to offer the following perspective.
> > >
> > > From the detector's viewpoint, a successful attack would need to reduce the number of green-list tokens in the scored region; otherwise the watermark signal remains intact and detection succeeds. This holds regardless of the attack mechanism:
> > >
> > > - **Paraphrasing** replaces tokens with alternatives that do not respect the green list
> > > - **Deletion** removes tokens (and their signal) entirely
> > > - **Rewriting** substitutes new content unaware of the green-list partition
> > >
> > > In each case, the mechanism by which detection is evaded is the removal of watermark signal. An attack that does not remove signal (i.e., the replacement tokens happen to preserve green-list membership) would not evade detection. Signal removal therefore captures the **core mechanism** shared by these attacks. Such type of attack is used as a benchmark in various litretures (Kuditipudi et al.(2023), Kirchenbauer et al. (2024), Cai et al. (2024), Liu et al.(2024)).
> > >
> > > Our framework provides exact characterization of how different detectors respond to this signal loss: the standard z-score retains $(1-f)$ of the signal, while WinMax-C retains $\sqrt{1-f}$.
> > >
> > >
> > > ### Additional results since the initial submission
> > >
> > > In response to the reviewer's concerns and those of other reviewers, we have extended the paper in several directions:
> > >
> > > - **Theoretical:** We introduced WinMax-C (a dual windowed detector) and proved that optimal parameters are shared across detectors ($O(1/\sqrt{n})$ convergence), with clean-calibrated parameters remaining asymptotically optimal under attack.
> > > - **Empirical:** We conducted cross-dataset quality analysis ($R^2$ regressions across 27 configurations), attack robustness experiments across 4 methods, 3 models, and 3 attack levels, as well as new experiments on Gemma-2 9B, confirming Pareto optimality of our method.
> > >
> > > We are grateful for the reviewer's time and effort throughout this discussion, which has helped us strengthen the paper considerably.
> > >
> > >
> > > Kuditipudi et al. (2023) https://arxiv.org/abs/2310.06356
> > >
> > > Kirchenbauer et al. (2024) https://arxiv.org/abs/2306.04634
> > >
> > > Cai et al. (2024) https://arxiv.org/abs/2403.13027
> > >
> > > Liu et al. (2024) https://arxiv.org/abs/2310.06356

---

### Official Review · Reviewer_1HAn · 2026-03-13

**Soundness:** 2
**Presentation:** 3
**Significance:** 3
**Originality:** 3
**Overall Recommendation:** 4
**Confidence:** 3

**Summary:**

This paper studies logit-based LLM watermarking and asks a focused question: how should the watermark hyperparameters be chosen to balance detectability and distortion? The authors argue that prior work gives limited guidance for principled tuning, so they develop a power-calibrated statistical framework that links watermark parameters directly to detection power and distortion, turning watermark design from heuristic trial-and-error into a constrained optimization problem. Concretely, they derive closed-form relationships for this trade-off, propose parameter-selection procedures under power or distortion constraints, and validate the framework on multiple models and datasets. Empirically, they report that the method consistently finds Pareto-optimal operating points and outperforms heuristic parameter tuning baselines in the detectability–quality trade-off.

**Compliance With Llm Reviewing Policy:**

Affirmed.

**Final Justification:**

My concerns have been addressed.

**Key Questions For Authors:**

Do you have evidence demonstrating that the KL-optimal point aligns with, or closely approximates, the true semantic quality-optimal point in most scenarios? Furthermore, Figure 5 only reports BLEU/ROUGE/BERTScore for the LFQA dataset; why was a systematic validation across all datasets not conducted? This is critical for determining whether the paper's use of the term "semantic distortion" is overstated.

**Limitations:**

Please see above

**Strengths And Weaknesses:**

Strengths：

1. This paper provides rigorous quantitative calibration of the detectability–distortion trade-off, addressing a key practical deployment challenge.

2.  The methodology follows a clear logical chain: from null/alternative hypotheses and $\gamma'$ mapping to power approximation, KL monotonicity, and constrained optimization.

3. It avoids pure abstraction by validating its core assumptions, specifically normality and $\gamma'$ fitting, rather than focusing solely on final TPR results.

Weakness:

1. The core derivation hinges on modeling the Next-Token Probability (NTP) vector as a uniform Dirichlet distribution. This is a significant mathematical simplification that fails to capture the sharp, context-dependent nature of real LLM distributions. The paper provides an "effective theory" but lacks a fundamental explanation for why such a coarse prior extrapolates well in practice.

2. To remain "conservative," the authors set the variance inflation factor $c=1$ in the main experiments. Since $c$ varies significantly across datasets and model domains, this approach reduces the "theoretical calibration" to a mere first-order approximation, failing to achieve truly precise power calibration.

3. Tests were limited to smaller models (e.g., GPT-2, OPT-125M), leaving the framework's performance on frontier large-scale models unverified.

---

> ### Author Rebuttal · Authors · 2026-03-30
>
> We thank the reviewer for the rigorous evaluation and for recognizing the clear logical chain of our methodology and its validation of core assumptions beyond final TPR results.
>
> ## W1: Dirichlet prior
>
> We agree that this is an important concern and appreciate the reviewer raising it. We address it in three parts: (1) the good approximation comes from **two-step concentration**; (2) under **mixture Dirichlet** (which can approximate any real NTP prior), our results extend with identical $\gamma'$; (3) using a **non-informative prior** is a fairness choice for algorithm analysis and comparison.
>
> - **Two-step concentration explains why the theory works.** The accuracy of our approximation is guaranteed by two concentration mechanisms.
>
>  **Vocabulary-level:** The green-list probability $Y = \sum_{i \in G} p_i$ concentrates around $\gamma$ for any NTP vector $p$, by the law of large numbers for random subset selection over a large vocabulary ($|V| \geq 50{,}000$). The watermarked green-token rate $g(Y) \to g(\gamma) = \gamma'$ follows from the smoothness of $g$.
>  **Sequence-level:** The detection statistic averages over $n$ independent token positions, each with an independent green-list partition, further reducing any residual per-token variance.
>
> The empirical validation ($R^2 \geq 0.98$ in Table 4 and alignment in Figure 4) confirms that the two-step concentration mechanism is sufficient for accurate approximation.
>
> - **Extension to mixture Dirichlet.** Dalal and Misra (2024, Theorem 2) show that any continuous bounded prior over multinomial probabilities can be approximated as a finite mixture of Dirichlet distributions. Indicating mixture Dirichlet is without loss of generality for practical NTP distributions. Our theory extends directly to this setting:  one replaces the uniform concentration parameter $\alpha_i = 1$ with component-specific $\alpha_i$, the concentration holds under mild regularity conditions, e.g., (1) $\sum_j \alpha_j$ grows with $|V|$, and (2) no single $\alpha_i$ dominates ($\max_i \alpha_i / \sum_j \alpha_j \to 0$). The mean is **invariant** regardless of the mixture weights, so our theoretical predictions carry over directly.
>
> - **Non-informative prior ensures fair comparison.** We chose not to pursue language-modeling-specific refinements. Since none of the baseline methods (GRID, DP, OPT) incorporate corpus-specific distributional information, assumption on corpus would introduce an advantage external to the calibration algorithm itself. Our measure ensures that our method's advantage comes purely from the optimization framework. We note that if additional modeling information (e.g., corpus-specific NTP statistics) becomes available, the concentration rate accuracy can be further improved.
>
>
>
>
>
> ## W2: Variance inflation factor
>
> We thank the reviewer for raising this point. After further investigation, we found that **$c$ does not affect the optimal parameter selection**: the solution $(\gamma^\ast, \delta^\ast)$ is identical for any $c > 0$, so setting $c=1$ does not compromise the calibration.
>
> - **The optimal $(\gamma^\ast, \delta^\ast)$ is invariant to $c$.** The power function has the form $\pi^\ast(\gamma, \delta) = \Phi(f(\gamma, \delta) / \sqrt{c})$, where $f(\gamma, \delta)$ is independent of $c$. Since $\Phi$ is strictly monotone:
>
> $$\arg\max_\gamma \Phi\left(\frac{f(\gamma, \delta)}{\sqrt{c}}\right) = \arg\max_\gamma f(\gamma, \delta)$$
>
> As we fix KL budget and maximize power, the Pareto-optimal $(\gamma^\ast, \delta^\ast)$ is **exactly the same** for any $c > 0$.
>
>
> - **Empirical $\hat{c}$ can refine power predictions** without changing the optimal parameters. We provide a data-driven procedure in Appendix C. Our calibrated values: $\hat{c} = 1.05$ (C4), $1.14$ (LFQA), $1.17$ (Wikipedia).
>
> ## W3: Model scale
>
> We appreciate the reviewer raising this point. To address this concern directly, we have conducted additional experiments on larger model (Gemma2-9b). The results confirm that our method **remains Pareto-optimal** at larger scales (see https://anonymous.4open.science/r/icml-2026-rebuttal-CD6D).
>
> ## Q1: Quality validation
>
> We agree that cross-dataset validation is important. Figure 5 in the paper reports only LFQA because the full 27 configurations (3 datasets $\times$ 3 models $\times$ 3 metrics) yield consistent conclusions. We include Pareto frontier figures (analogous to Figure 5) for all dataset-model combinations in (https://anonymous.4open.science/r/icml-2026-rebuttal-CD6D).
>
>
> Additionally, we fitted linear regressions of BLEU, ROUGE, and BERTScore against empirical KL divergence using all experimental data:
>
> | Metric | R² |
> |--------|-----|
> | BLEU | 0.566 |
> | ROUGE | 0.870 |
> | BERTScore | 0.791 |
>
> KL explains a large amount of variety in other scores. This confirms that **minimizing KL divergence is an effective proxy for preserving sequence-level quality**.
>
> [1] Dalal and Misra (2024) https://arxiv.org/abs/2402.03175

---

> > ### Author Rebuttal · Reviewer_1HAn · 2026-04-01
> >
> > My concerns have been addressed, and I have raised my score accordingly.

---

> > > ### Author Response · Authors · 2026-04-01
> > >
> > > We thank the reviewer for the positive assessment and are encouraged to hear that our previous responses were satisfactory. Your constructive input has been instrumental in strengthening this paper, and we truly appreciate your time and guidance throughout the review process.

---

### Official Review · Reviewer_HY4y · 2026-03-17

**Soundness:** 2
**Presentation:** 3
**Significance:** 2
**Originality:** 2
**Overall Recommendation:** 4
**Confidence:** 4

**Summary:**

The paper proposes a power-calibrated statistical framework for logit-based LLM watermarking, modeling detection as a hypothesis testing problem and deriving explicit relationships between watermark strength, detection power, and distortion. This formulation enables principled parameter selection that replaces heuristic tuning with optimization under power or distortion constraints. Experiments demonstrate that the proposed framework identifies Pareto-optimal watermark configurations across models and datasets.

**Compliance With Llm Reviewing Policy:**

Affirmed.

**Final Justification:**

I thank the authors for their rebuttal, which has addressed my main concern. Therefore, I would like to increase my score.

**Key Questions For Authors:**

1. Is the disotrtion budget defined over the entire sequence or on each token?
2. Does one need to solve the optimization problem (2) at each token?

**Limitations:**

The paper should also discuss both the potential risks of watermarking.

**Strengths And Weaknesses:**

### Strengths
- Presentation: The paper is in general well organized.
- Significance: The paper studies watermarking for LLM-generated content, an increasingly important topic for AI safety and security. In addition, the authors attempt to provide theoretical understanding beyond purely empirical algorithm design.
- Originality: The paper jointly optimizes the green list portion and perturbation strength so as to covert the distortion budget into detectability gains, which has not been fully studied before.


### Weaknesses
1. The hypothesis testing problem is not well explained. What is the probability measure $\mathbb{P}$ under $H_0$ and what is $\mathbb{P}_\delta$ under $H_1$? Furthermore, this problem is defined only on the token level. How about on the sequence level?
2. The hypothesis testing problem is only on the statistic $I_i$, which is surrogate formulation is not the true problem.
3. Under Theorem 3.8, $\pi^*(\gamma,\delta)$ is not defined in the previous parts. Why is it called the power? How does Theorem 3.8 imply this expression? Why is it optimal?
4. Minor: Impact Statement should not have section number.

---

> ### Author Rebuttal · Authors · 2026-03-30
>
> We thank the reviewer for the careful reading and for recognizing the originality of jointly optimizing $(\gamma, \delta)$ to convert the distortion budget into detectability gains.
>
> ## W1: Hypothesis Testing
>
> We thank the reviewer for raising this question; we will clarify these notations.
>
> - $\mathbb{P}$ abstracts the LLM as a probability measure over token sequences. $\mathbb{P}(x_i = k \mid x_{<i})$ is the next-token probability when the context is $x_{<i}$, i.e., the LLM's original output distribution.
> - $\mathbb{P}\_\delta$ is the watermarked counterpart: $\mathbb{P}\_\delta(x_i = k \mid x_{<i})$ is the distribution after adding logit bias $\delta$ to green-list tokens (Eq. in line 077--079).
> - **Notational convention:** Strictly speaking, the latter should be conditioned on the green list $G_i$. We adopt the shorthand because $G_i$ is fully determined by $x_{<i}$ and the secret key (i.e., $G_i = G(x_{<i}, \xi)$ is not an additional source of randomness). We will make this explicit in the revision.
>
>
>
> **Sequence level:** Our analysis proceeds token-first, then aggregates. Sections 3.1--3.2 characterize the per-token signal ($\gamma$ and $\gamma'$). Section 3.3 aggregates over $n$ tokens: $S_n = \sum_{i=1}^n I_i \sim \text{Binomial}(n, \gamma)$ under $H_0$, and the CLT gives the closed-form power $\pi^\ast(\gamma, \delta)$ at the sequence level (Theorem 3.8). We will discuss this token-to-sequence progression more clearly in the revision.
>
> ## W2: Formulation
>
> We appreciate this concern. The green-list indicator $I_i = \mathbf{1}\{x_i \in G_i\}$ captures the entire watermark signal at each token position. At the sequence level, their sum $S_n = \sum_{i=1}^n I_i$ is the **sufficient statistic** of the watermarking problem. $S_n$ captures the full watermark signal, so our formulation addresses the sequence-level detection problem directly, not a surrogate of it.
>
> ## W3: $\pi^\ast(\gamma, \delta)$ not defined
>
> We thank the reviewer for this observation. We will clarify the presentation in the revision:
>
> - **"Power"** is standard statistical terminology: the probability of correctly rejecting $H_0$ when the watermark is present (Type II error $= 1 - \pi^\ast$).
> - **Definition:** $\pi^\ast(\gamma, \delta) := \mathbb{P}\_\delta(\text{reject } H_0)$ is the detection power for a watermarked sequence of length $n$. By Theorem 3.8, the z-test statistic converges to a normal distribution under $H_1$, yielding the closed-form expression $\pi^\ast = \Phi\left(\frac{\sqrt{n}(\gamma'-\gamma) - z_{1-\alpha}\sqrt{\gamma(1-\gamma)}}{\sqrt{c\gamma'(1-\gamma')}}\right)$, where $\Phi$ is the standard normal CDF, $\gamma'$ is the watermarked green-token rate (Theorem 3.5), and $z_{1-\alpha}$ is the significance threshold.
> - **"Optimal"** refers to the fact that $\pi^\ast$ **directly characterizes detection power**, rather than relying on surrogate measures (e.g., $\Delta P$ in prior work). Surrogates introduce approximation error, which can cause the optimizer to over- or under-allocate KL budget, leading to suboptimal Pareto trade-offs.
>
> We will add a formal definition of $\pi^\ast$ before Theorem 3.8 and clarify its optimality in the revision.
>
> ## W4: Minor formatting
>
> We appreciate the reviewer's careful reading. We will remove the section number in the revision.
>
> ## Q1: Distortion budget
>
> We thank the reviewer for this question. The distortion budget $K_0$ in Eq. (2) is the **per-token expected KL divergence** $D_\text{KL}(\gamma, \delta)$. Since the total sequence-level KL is the sum of token-level KL divergences (as stated in Section 3.4, following Cai et al., 2024), and $(\gamma, \delta)$ are fixed for the entire generation, the per-token KL is constant:
>
> $$\frac{1}{n} D_\text{KL}^\text{sequence} = D_\text{KL}(\gamma, \delta) = K_0$$
>
> ## Q2: Optimization at each token?
>
> We thank the reviewer for this question. **Optimization (2) does not need to be solved at each token; it is solved once before generation.**
>
> - **Model-agnostic by necessity:** The watermark detector only observes the generated text and cannot recover the underlying model or its per-token probabilities. Calibration must therefore operate at the macroscopic level rather than tracking per-token fluctuations.
> - **One-time offline computation (our core contribution):** By deriving closed-form expressions for both $D_\text{KL}(\gamma, \delta)$ and $\pi^\ast(\gamma, \delta)$ that depend only on $(\gamma, \delta)$, our framework reduces parameter selection to a single offline optimization, applied uniformly throughout generation.
>
> We will state this more explicitly in the revision.
>
> ## Limitations discussion
>
> We will expand the limitations section to discuss potential risks of watermarking (e.g., false accusations, adversarial removal) and the framework's current scope.
>
> [1] Cai et al., (2024) https://arxiv.org/abs/2403.13027

---

> > ### Author Rebuttal · Reviewer_HY4y · 2026-04-02
> >
> > Thank you for your response. However, some of my concerns remain under-addressed.
> >
> > - `R-W1`: Do you consider a sentence-level hypothesis testing in Section 3.3? If so, what would be the formulation?
> > - `R-W2`: The true and original hypothesis testing problem is:  $H_0: x_i \text{unwatermarked, i.e., } x_i \sim \mathbb{P}$; $H_1: x_i \text{watermarked, i.e., } x_i \sim \mathbb{P_\delta}$. This is not equivalent to the HT problem in Section 3.1. Furthermore, it has not been proven that S_n is a sufficient statistic for the true hypothesis testing problem, and I am skeptical that it is.
> > - `R-Q1`: It means that the distortion at the sequence level grows with the sequence length then. Is that reasonable in practice? How to control the sentence quality in this sense?

---

> > > ### Author Response · Authors · 2026-04-06
> > >
> > > We thank the reviewer for the continued engagement and the important follow-up questions.
> > >
> > > ### R-W1: Sentence-level hypothesis testing
> > >
> > > Yes, the sentence-level formulation is presented in Section 3.3. Let $S_n = \sum_{i=1}^n I_i$, where $\gamma = \mathbb{P}(x_i \in G_i)$ under the null (Lemma 3.2) and $\gamma' = \mathbb{P}_\delta(x_i \in G_i)$ under the alternative (Theorem 3.5). By CLT (Theorem 3.8, Section 3.3), the sequence-level behavior is:
> > >
> > > Under $H_0$: $\sqrt{n}(S_n/n - \gamma) \xrightarrow{d} \mathcal{N}(0, \gamma(1-\gamma))$
> > >
> > > Under $H_1$: $\sqrt{n}(S_n/n - \gamma') \xrightarrow{d} \mathcal{N}(0, c\gamma'(1-\gamma'))$
> > >
> > > As a result, the detection power $\pi^\ast(\gamma, \delta)$ admits a closed-form expression (Theorem 3.8).
> > >
> > > ### R-W2: Problem formulation
> > >
> > > We appreciate this important question. We argue that our hypothesis testing formulation ($H_0: \gamma$ vs $H_1: \gamma'$) is not a surrogate but rather the maximally informative formulation available at detection time.
> > >
> > > - **$H_0$ on $V^n$ is not well-posed.** We agree that $H_0$ is "text not watermarked," but this does not imply $x_1^n \sim \mathbb{P}$: it must also cover human-written text, which is not generated in an autoregressive fashion and admits no tractable parametric characterization. Without a parametric $H_0$, the sufficient statistic is the entire sample $x_1^n$. Since the distribution under $H_0$ is unknown, threshold calibration, FPR control, and power analysis become difficult.
> > >
> > > - **All unwatermarked text shares $\mathbb{P}(x_i \in G_i) = \gamma$.** Despite the diversity of sources under $H_0$, the green list $G_i$ is determined by a cryptographic hash of $(x_{<i}, \xi)$, independent of the text source (Assumption 3.1, Lemma 3.2). Through the indicator $I_i = \mathbf{1}\{x_i \in G_i\}$, the composite $H_0$ collapses into a simple hypothesis: $H_0: \mathbb{P}(x_i \in G_i) = \gamma$. This is well accepted in related litreture, under $H_0$, $S_n = \sum I_i$ follows a Binomial distribution, for which $S_n$ is the sufficient statistic.
> > >
> > > - **Exact characterization of $H_1$.** Intuitively $\gamma' > \gamma$ under watermarking, but we provide the **exact characterization** $\gamma'(\gamma, \delta)$ (Theorem 3.5). This is an if-and-only-if relationship: the watermark mechanism yields $\gamma'$ (forward), and producing $\gamma' > \gamma$ without the secret key $\xi$ is computationally infeasible (reverse). Combined with point 2 above ($H_0: \gamma$), our hypothesis testing formulation $H_0: \gamma$ vs $H_1: \gamma'$ is a complete characterization of the original detection problem. With the token-level problem precisely specified, we extend to the sequence level via CLT (Theorem 3.8), enabling analytical derivation of power, distortion, and optimal parameters.
> > >
> > > - **Information utilization.** At detection time, the watermark-related information available to the detector is the green lists and the derived green-token counts. Our hypothesis testing formulation utilizes this information fully.
> > >
> > > In summary, our hypothesis testing formulation is not a surrogate but a grounded utilization of the information available.
> > >
> > >
> > > ### R-Q1: Sequence-level distortion
> > >
> > > We thank the reviewer for this insightful question, which touches on a fundamental aspect of watermark design. The total $D_\text{KL}^\text{sequence} = n \cdot D_\text{KL}(\gamma, \delta)$ does grow linearly with $n$. We offer the following observations.
> > >
> > > - **Distributional change and signal are inseparable.** The watermark signal exists precisely where the distribution has been modified; a position with no distributional change carries no watermark and is undetectable. Signal injection is therefore present across the entire generation. That said, Kirchenbauer et al. (2024b) observe that the quality impact is marginal with reasonable parameters, consistent with natural language's substantial redundancy (Shannon, 1951).
> > >
> > > - **Our framework optimizes the trade-off.** Given that per-token distortion is unavoidable, the natural question is how much detection power can be gained per unit of distortion. Our framework addresses this by finding $(\gamma^\ast, \delta^\ast)$ that maximizes power $\pi^\ast$ for a given per-token budget $K_0$, with a closed-form solution.
> > >
> > > - **Longer text allows weaker watermarking.** In many practical tasks (e.g., long-form generation, document drafting), producing long sequences is common. For such scenarios, the signal accumulation over many tokens means a strong per-token watermark is unnecessary. One can adopt the dual formulation (Section 4): fix a target power and minimize KL Divergence. The $\sqrt{n}$ factor in $\pi^\ast$ amplifies even a small per-token signal over many tokens, so longer sequences naturally require weaker per-token watermarking for the same detectability.
> > >
> > > We will clarify the relationship between sequence-level KL and per-token control more explicitly in the revision.
> > >
> > > Shannon (1951) https://www.princeton.edu/~wbialek/rome/refs/shannon_51.pdf

---

### Decision · Program_Chairs · 2026-04-30

**Decision:**

Accept (regular)

**Comment:**

This paper develops a statistical framework for calibrating logit-based LLM watermarks, explicitly linking watermark parameters to detection power and distortion so that parameter selection can be done systematically rather than heuristically. The reviewers generally viewed this as a technically solid and timely paper on LLM watermarking, with a clear practical goal. Strengths highlighted across reviews included the paper’s strong theoretical structure, explicit optimization view, and empirical validation of the theory. The main concerns were about clarity and scope: one reviewer questioned the statistical hypothesis-testing formulation and whether the green-token statistic is truly sufficient; another raised concerns about the simplifying Dirichlet prior, and limited validation on larger models; and a third noted that the framework is specialized to KGW-style watermarking, and does not fully study robustness under editing attacks.

The rebuttal was generally effective and moved the discussion in a positive direction. The authors clarified the token-to-sequence testing formulation, explained the role of the detection power expression, answered questions about per-token versus sequence-level distortion, and argued that optimization is done once offline rather than at each token. They also added evidence on larger models, provided further justification for the Dirichlet-based approximation and KL-based quality proxy, and offered additional discussion and experiments on robustness under signal-removal attacks. Two reviewers stated that their concerns were addressed, and updated their final justification positively. Overall, the rebuttal strengthened the paper and supports an accept recommendation, while leaving some limitations in breadth and deployment realism.